# Exploring Variability in Climate Change projections on the Nemunas River and Curonian Lagoon: coupled SWAT and SHYFEM modeling approach

Natalja Čerkasova [1,2]‡, Jovita Mėžinė [1]‡, Rasa Idzelytė [1]‡, Jūratė Lesutienė [1], Ali Ertürk [1,3], Georg Umgiesser [1,4]

[1] Marine Research Institute, Klaipeda University, Klaipeda 92294, Lithuania
[2] Texas A&M AgriLife Research, Blackland Research and Extension Center, Temple, TX 76502, USA
[3] Department of Inland Water Resources and Management, Istanbul University, Istanbul 34134, Turkey
[4] CNR – National Research Council of Italy, ISMAR – Institute of Marine Sciences, Venice 30122, Italy

*Correspondence to*: Jovita Mėžinė, jovita.mezine@ku.lt

‡Equally contributed: Natalja Čerkasova, Jovita Mėžinė, Rasa Idzelytė

**Abstract.** This study advances the understanding of climate projection variabilities in the Nemunas River, Curonian Lagoon, and southeastern Baltic Sea continuum by analyzing the output of a coupled ocean and drainage basin modeling system forced by a subset of climate models. A dataset from a downscaled high-resolution regional atmospheric climate model driven by four different global climate models was bias-corrected and used to set up the hydrological (SWAT) and hydrodynamic (SHYFEM) modeling system. This study investigates the variability and trends in environmental parameters such as water fluxes, timing, nutrient load, water temperature, ice cover, and saltwater intrusions under Representative Concentration Pathway 4.5 and 8.5 scenarios. The analysis highlights the differences among model results underscoring the inherent uncertainties in projecting climatic impacts, hence highlighting the necessity of using multi-model ensembles to improve the accuracy of climate change impact assessments. Modeling results were used to evaluate the possible environmental impact due to climate change through the analysis of the Coldwater fish species reproduction season. We analyze the duration of cold periods (<1.5°C) as a thermal window for burbot (*Lota lota* L.) spawning, calculated assuming different climate forcing scenarios and models. The analysis indicated coherent shrinking of the cold period and presence of the changepoints during historical and different periods in the future, however, not all trends reach statistical significance, and due to high variability within the projections, they are less reliable. This means there is a considerable amount of uncertainty in these projections, highlighting the difficulty in making reliable climate change impact assessments.

## 1 Introduction

A river-lagoon-sea continuum is a very complex system that forms a unique and vulnerable environment providing a broad spectrum of the ecosystem services (Kaziukonyte et al., 2021; Inácio et al., 2018) and plays an important socioeconomic role. On the larger scale the climate change impacts are extensively analyzed and already showed that

the coastal zone will be impacted by the global warming, sea level rise, by altering of a freshwater runoff, frequency and intensity of coastal storms, precipitation and nutrients patterns (Viitasalo and Bonsdorff, 2022; Lu et al., 2018). Modeling becomes an important tool to project climate change impact with the focus on the intensity and direction of future changes. However, there are a lot of uncertainties regarding the trends and projected impacts due to climate change (IPCC, 2013). The uncertainties and variations of projected future scenarios emerge due to unknowns in global or regional climate models (GCMs, RCMs), proposed scenarios (RCPs), or statistical techniques used for data preparation. Therefore, uncertainty analysis is commonly used to quantify the possible discrepancies between the projections and their impacts on possible future changes. There is a wide variety of studies focused on quantification of climate projection uncertainties around the world, including Lithuania (e.g., Chen et al., 2022; Song et al., 2020; Akstinas et al., 2019). Most of these studies analyze solely hydrological changes due to meteorological input.

The uncertainty in climatic studies arises from various factors, as highlighted by Foley (2010). One key factor is the scenario used as the basis for climatic projections. These scenarios range from significantly reduced $CO_2$ emissions to business-as-usual cases, i.e., continuation of high emissions-based economic growth, leading to vastly different climate trajectories (Latif M., 2011, Taylor et al., 2012). Even if the underlying assumptions are consistent, the climate models used are handling the physics differently leading to different results of the key parameters (Lehner et al., 2020). Apart from the atmospheric models, there is also a variety of ocean models, for example NEMO (Madec et al., 2016), POM (Mellor, 2004), ROMS (Shchepetkin and McWilliams, 2005), MITgcm (Marotzke et al., 1999), SHYFEM (Umgiesser et al. 2004) and others, that have to be considered. All of these models have different discretization, resolution, and representation of the physics modeled. Drainage basin models depend crucially on the changing land use of the basin (Wang et al., 2012, Lin et al., 2015, Waikhom et al., 2023), with subsequent effects on downstream coastal ecosystems.

The development of integrated modeling tools is a high-priority task to support the management of the ecosystems at the land-sea interface, prone to both the riverine effects and sea level rise. This study is a continuation of the previously published paper by Idzelytė et al. (2023a) where the framework of coupled hydrological and hydrodynamic models was used to explore the future climate scenarios based on the ensemble mean values for the Nemunas River watershed, Curonian Lagoon, and Baltic Sea continuum. Here, we explore a subset of the possible variation space. We look at different scenarios computed by different climate models, using only one ocean model (Umgiesser et al., 2004) and one drainage basin model (Čerkasova et al., 2018). This allows us to come up with a reasonable estimate of the variability of climate projections and its impact on the hydrology and its application to the ecological evaluation of the studied Nemunas River basin, the Curonian Lagoon, and the southeastern Baltic Sea system as a whole.

It is expected that explicit analysis of the climate scenarios will help decision-makers in the development of climate change adaptation and mitigation strategies as well as adjustment of water quality management, and achievement of regional nutrient policy goals and measures. The level of uncertainty is crucial in the decision-making process, therefore we aim to test model averaging (Idzelytė et al. 2023a) vs. the ensemble method, where we combine the results of several models to form an ensemble projection. The diversity in projections among the ensemble components may reveal the level of variability and aid in combining agriculture nutrient runoff policies with climate mitigation policies that involve integrating strategies to address both issues simultaneously.

Climate prediction uncertainty has important implications for the conservation efforts of endangered or vulnerable
species, as meteorological-hydrological factors play a primary role in shaping species habitat conditions, life cycle
completion, spread, and survival. In addition to the variability in projections for the region, we specifically tackle the
question of how much the imposed changes could be reflected in ecosystem function and habitat conditions for the
species. As the response of climate forcing is most pronounced in water temperature, we selected the stenotherm
species burbot (*Lota lota* L.). As a coldwater fish species, the burbot is particularly sensitive to changes in thermal
habitat availability (Harrison et al., 2016) and suffers severe declines throughout its distribution range worldwide
(Stapanian et al., 2010). Evaluating the impact of climate change on spawning habitats is essential for projecting the
future status of the vulnerable burbot population in the Curonian Lagoon.
**2 Materials and methods**
**2.1 Study area**
Our study site is a large transboundary basin - coastal lagoon - sea system: Nemunas River basin, the Curonian Lagoon,
and the southeastern Baltic Sea. The Curonian Lagoon is a shallow estuarine lagoon located in Lithuania and Russian
Federation's territory and connected to the south-eastern Baltic Sea through the narrow Klaipėda Strait (Fig. 1). The
lagoon covers an area of 1584 km$^2$, with its widest section stretching up to 46 km in the southern part. Conversely, in
the northernmost part (Klaipėda Strait), it narrows down to approximately 400 m wide. The drainage area of the
Curonian Lagoon covers 100 458 km$^2$, of which 48% lies in Belarus, 46% in Lithuania, and 6% in the Kaliningrad
oblast. Previous hydrodynamic modeling studies revealed that the lagoon consists of two different regions from the
water exchange point of view, a transitional region at the northern part of the lagoon and a stagnant southern region
which has a considerably higher water residence time. The predominant flow of water is from the south to the north
discharging approximately 23 km³ per year into the Baltic Sea.

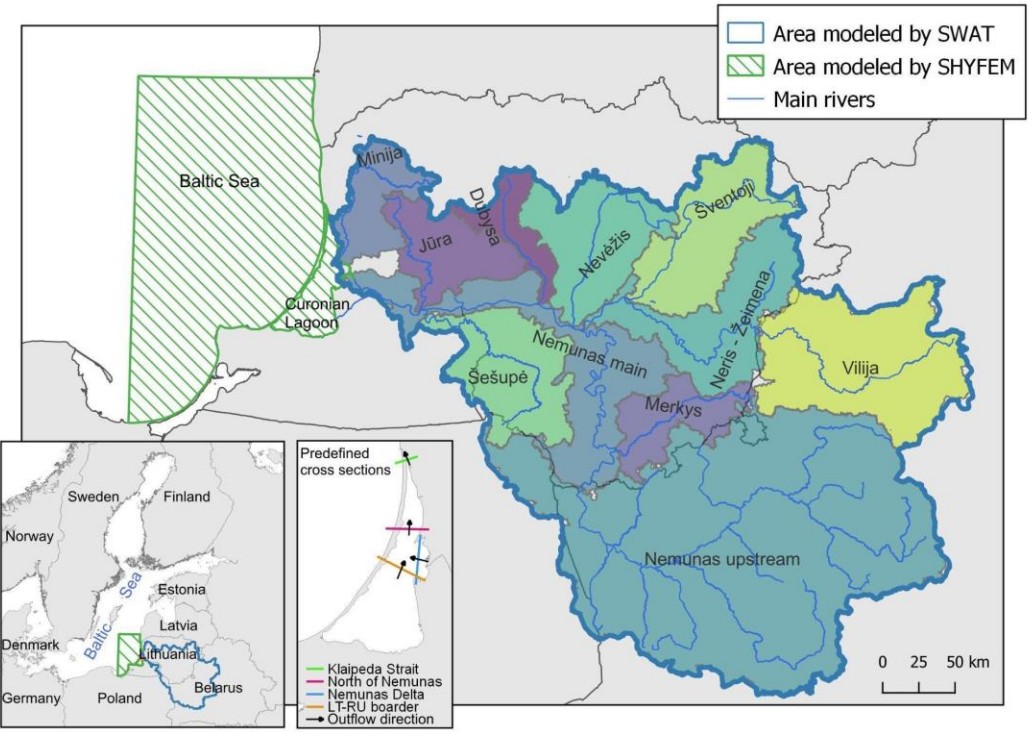

**Figure 1. Location of the Curonian Lagoon and Nemunas River Watershed**.

The largest river that discharges into the Curonian Lagoon is the Nemunas River, which together with the Minija River brings about 95% of the total riverine input to the lagoon (Zemlys et al., 2013). Both rivers enter the lagoon in the middle of the eastern coast. The average annual discharge of the Nemunas River is 22-24 km$^3$ (Umgiesser et al., 2016) and exhibits a strong fluctuating seasonal pattern, peaking with snowmelt during the flood season in February-April. Due to discharge from the Nemunas River and other smaller rivers, the southern and central portions of the lagoon are considered to be freshwater.

The Curonian Lagoon and Nemunas Delta area both include protected territories with various statuses: biosphere polygons, reserves, Natura 2000 (Special Protection Areas (EC Birds Directive), Sites of Community Importance (EC Habitats Directive)) and Ramsar List site (List of Wetlands of International Importance) (Kaziukonyte et al, 2022). The Curonian Lagoon and Nemunas Delta are the most important areas for commercial fishing in Lithuania, contributing about 95-98% of the total inland fishery (Ivanauskas et al, 2022). Bream (*Abramis brama* L.), pikeperch (*Sander lucioperca* L.), and smelt (*Osmerus eperlanus* L.) are the main commercial fish species in the lagoon. In the context of climate change, coldwater species like burbot (*Lota lota* L.) are particularly sensitive. They rely on low water temperatures during winter to initiate the spawning season.

**2.2 Modeling system**

Due to limitations in current technology and tools, accurately representing the entire Nemunas River basin, Curonian Lagoon, and southeastern Baltic Sea system at high resolution with a single tool is impossible. As a result, we divided the area and utilized various modeling tools suited for specific purposes, which were coupled together. The modeling

system that consists of two main models and numerous utilities mostly developed to transfer the outputs from one
model as inputs to other models are summarized in Fig. 2. The system is characterized by two pivotal models: 1) the
hydrological Soil and Water Assessment Tool (SWAT) model, and 2) the hydrodynamic Shallow water
HYdrodynamic Finite Element Model (SHYFEM). The SWAT and SHYFEM models depict main water flow
dynamics in a Nemunas River watershed-Curonian Lagoon-Baltic Sea continuum.

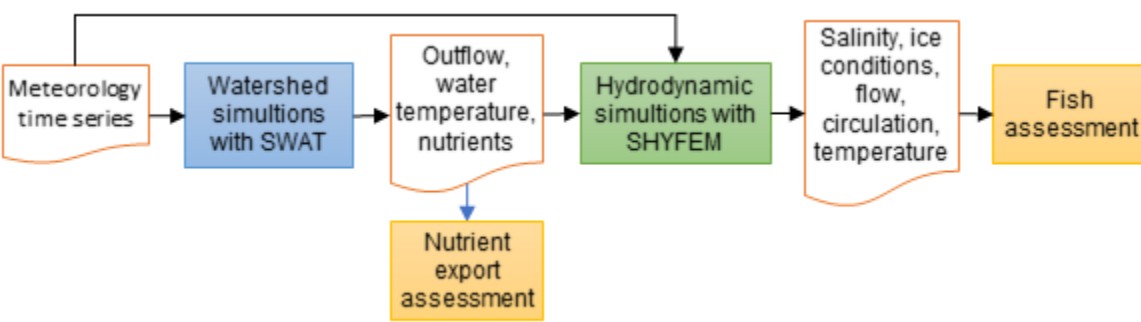

**Figure 2. Hierarchical structure of the modeling system.**
The Nemunas River watershed is modeled using the SWAT (Neitsch et al., 2009) which is widely used to simulate
hydrological processes and water quality of watersheds. This model was developed, calibrated, and validated for the
Nemunas River basin in previous studies (Čerkasova et al., 2021, 2019, 2018). The SWAT is a comprehensive tool
that requires numerous model inputs for hydrological parameterization and watershed characterization. The inputs can
differ based on modeling demands and the topographic characteristics of the region. For the Lithuanian part of the
watershed regional high-resolution data (such as the Digital Elevation Model, land use, soil, stream network, reservoir
information) were gathered from the governmental institutions in Lithuania (see Table 1). Where data was not
available, European or global datasets were used, in combination with information found in relevant literature. To
ensure accuracy, we manually digitized the stream network (used as the burn-in layer for watershed delineation and
routing information), reservoir and pond geometry (used to identify the standing waterbody location and for parameter
calculation), and major forest outlines (used to correct the land use layer).
Due to the large basin area and heterogeneity in topography and land management in the region, the entire watershed
covering all the Nemunas River Basin was split into separate SWAT sub-models, each representing a sub-watershed
of the main Nemunas River branch. Furthermore, to achieve better parametrisation a separate sub-model represents
the Nemunas and all smaller tributaries situated in the Belarus and Poland territories. The outcome of division into the
sub-models produced the following configuration:
●  1 sub-model in the Belarus territory (Neris in Lithuanian, Вíлія in Belarus);
●  2 transboundary watersheds:
○  Sesupe (Šešupė in Lithuanian, Шешупе in Russian, Szeszupa in Polish);
○  Nemunas upstream (Неман in Russian and Belarus);
●  7 sub-models with more than 95% of the territory in the boundary of Lithuania or entirely situated in
Lithuania:

○    Minija;

○    Merkys;

○    Jura (Jūra in Lithuanian);

○    Dubysa;

○    Sventoji (Šventoji in Lithuanian);

○    Nevezis (Nevėžis in Lithuanian);

○    Neris- Zeimena (Neris-Žeimena in Lithuanian);

●    1 sub-model, - the Nemunas main branch - discharging into the Curonian Lagoon.
A total of eleven sub-models were built, subdivided into subbasins (in total 9012), which were further subdivided into
Hydrological Response Units (in total 148 212 HRUs), configured, connected, and parametrized. The concept of the
eleven SWAT models that are represented as sub-models for the entire study area is given in Figure 1 (denoted as
separate colors of the watershed). These models can be used individually or, as in this study, in a framework, where
the upstream sub-models provide the input information to the downstream areas. Outputs from the main outlets on the
Nemunas and Minija rivers were used as boundary conditions for the hydrodynamic model.
Calibration and validation of each sub-model were conducted manually by adjusting parameters linked to specific
processes. The multisite calibration process followed an approach typical for hydrological models (Daggupati et al.,
2015; Feyereisen et al., 2007). Calibration began with the upstream regions, followed by the downstream areas,
focusing on flow, sediment, total nitrogen (TN), and total phosphorus (TP). This methodology was applied both to
individual sub-models and the overall modeling framework. Details can be found in Čerkasova et. al (2021).
The hydrodynamics of the Curonian Lagoon and the southeastern Baltic Sea were simulated using the open-source
shallow water hydrodynamic finite element model SHYFEM, accessible at https://github.com/SHYFEM-model/ (last
accessed on 28 November 2023). The model uses an unstructured grid (finite elements) to discretize the studied basin
(Curonian Lagoon and part of the Baltic Sea). The use of finite elements is crucial in order to simulate the narrow
connection of the lagoon with the sea (Klaipeda Strait). However, it varies from 250 m close to the Klaipeda Strait to
up to 2.5 km in the central part of the lagoon and up to 10 km in the Baltic Proper. The atmospheric forcing has been
interpolated directly from the regular grid of the regional climate model data to the finite element nodes by bi-linear
interpolation. Lateral boundary conditions have been taken from Copernicus data and interpolated onto the finite
element grid (water levels, T, S). The COARE3.0 module is used for bulk formulation. The model solves shallow
water equations and, in this study, the 2D version of the model was used. The SHYFEM simulates key physical
variables such as circulation, waves, water level, temperature, and salinity fields that are needed to characterize the
water matrix. To compute the water fluxes across the sides of the elements, first the conservation of mass in the finite
volume around a node that is guaranteed by the continuity equation is used. The fluxes over the lines delimiting the
finite volume element per element are made divergence free by subtracting the storage of water inside the node. With
these finite volume fluxes the fluxes over the element sides are computed. This tool has been applied to a large number
of lagoons around Europe. Details can be found in (Idzelytė et al., 2020, Umgiesser et al., 2016, Zemlys et al., 2013,
Umgiesser et al. 2014, and Umgiesser et al. 2004).

## 2.3 Data

Our modeling system incorporates different input data, varying according to the specific model utilized - either hydrological or hydrodynamic (as outlined in Table 1). Given that this study follows the research conducted by Idzelytė et al. (2023a), to delve into the specifics of the input data utilized in our study we refer the reader to their previously published work.

| | Input data type | Source |
|---|---|---|
| **Hydrological** | Digital Elevation Model (DEM) | National Land Service under the Ministry of Agriculture of Republic of Lithuania<br>The Shuttle Radar Topography Mission (SRTM) 1 Arc-Second Global |
| | Land use and management Data | National Land Service under the Ministry of Agriculture of Republic of Lithuania<br>WaterBase project database<br>Corine landcover 2012<br>Lithuanian Environmental Protection Agency<br>Eurostat<br>National Statistical Committee of the Republic of Belarus<br>Ministry of natural resources and environmental protection of the Republic of Belarus |
| | Hydrologic grid | National Land Service under the Ministry of Agriculture<br>The Ministry of Agriculture of the Republic of Lithuania<br>Reports of Belarus government agencies, fishing enthusiasts portals<br>Manual digitization using satellite data |
| | Soil maps | National Land Service under the Ministry of Agriculture<br>Lithuanian Soil atlas |
| | Observed discharge and nutrient data | Lithuanian Hydrometeorological Service<br>Lithuanian Environmental Protection Agency |
| | Crop yield | Lithuanian Statistical Yearbook<br>National Statistical Committee of the Republic of Belarus |
| | Daily precipitation and air temperature (min/max) | Cordex RCA4 data after bias correction |
| **Hydrodynamic** | Water level, temperature, and salinity | RCA4–NEMO model developed by the Rossby Centre and the oceanographic research group at the Swedish Meteorological and Hydrological Institute (SMHI). The bias correction was done by simply adding the difference between the average values of CMEMS and RCA4–NEMO data (Lenderink et al., 2007) |
| | Bathymetry | The Leibniz Institute for Baltic Sea Research Warnemünde (IOW) |
| | Ice thickness | ESIM2 model |
| | Meteorological forcing (wind, pressure, air temperature, solar radiation, cloud cover, precipitation) | Cordex RCA4 data after bias correction |
| **Validation** | Precipitation and Air temperature | Lithuanian Hydrometeorological Service (1993-2005), 18 meteorological stations, which are scattered throughout the Republic of Lithuania |
| | Water level, temperature and salinity | Copernicus Marine Environment Monitoring Service (CMEMS) Baltic Sea Physics Reanalysis product data (1993–2005) |

**Table 1. Input and validation data types for the hydrological and hydrodynamic modeling system and their respective sources.**

Both hydrological and hydrodynamic models were run using the same bias corrected future meteorological forcing data described in Table 2. Data were obtained from CORDEX (Coordinated Regional Downscaling Experiment) scenarios for Europe, employing the Rossby Centre high-resolution regional atmospheric climate model (RCA4). This involved four sets of simulations (downscaling) driven by four global climate models. The datasets are spanning the

historical period of 1970–2005 and the projection period of 2006–2100. Projections are based on two Representative
Concentration Pathway (RCP) scenarios, specifically RCP4.5 and RCP8.5 of the Coupled Model Intercomparison
Project Phase 5 (CMIP5). The bias correction was conducted by applying the climate data bias correction tool (Gupta
et al., 2019). The ice thickness data utilized in our study were derived using the ESIM2 model (Tedesco et al., 2009,
Idzelytė and Umgiesser, 2021). This model was run independently as a standalone system, and the resulting output
time series were integrated into our hydrodynamic modeling framework as surface boundary input data. This approach
allowed us to accurately incorporate ice thickness dynamics into our simulations, enhancing the overall reliability of
our model during the ice season. A detailed description of all the data sets used for this study can be found in Idzelytė
et al. (2023a), while the results derived from the modeling system can be found and accessed in the open-access
Zenodo database (https://doi.org/10.5281/zenodo.7500744, Idzelytė et al. (2023b)).

| Abbreviation | Model | Institution |
|---|---|---|
| ICHEC | EC-Earth - A European community Earth System Model | Irish Centre for High-End Computing |
| IPSL | IPSL-CM5A-LR - Institut Pierre Simon Laplace - Earth System Model for the 5th IPCC report: Low resolution | The Institute Pierre-Simon Laplace |
| MOHC | HadGEM2-ES - Hadley Global Environment Model 2 - Earth System | Met Office Hadley Centre |
| MPI | MPI-ESM-LR - Max-Planck-Institute Earth System Mode: Low resolution | Max Planck Institute for Meteorology |

**Table 2. Meteorological forcing data sources for the hydrological and hydrodynamic modeling system.**
**2.4 Analysis methods**
**2.4.1 Investigation of hydrological and hydrodynamic model results**
The analysis was done for the environmental parameters corresponding to our preceding study (Idzelytė et al., 2023a).
These include air temperature, precipitation, Nemunas River discharge, water inflow and outflow from the lagoon at
different locations such as Klaipėda Strait, North of Nemunas, Nemunas Delta, and along the Lithuanian-Russian (LT-
RU) border. In this analysis, we maintained the inflow and outflow categories as in our previous study (Idzelytė et al.,
2023a). We analyzed the data by computing the 10-year moving average using yearly average fluxes, this way ensuring
an accurate representation of water flux dynamics throughout the study period. Water temperature and water level
were evaluated for the Southeast (SE) Baltic Sea and Curonian Lagoon. Saltwater intrusions ($>2$ g kg$^{-1}$) were assessed
in Juodkrantė, approximately 20 km south of Klaipėda Strait. Information on ice cover in the Curonian Lagoon
encompasses the season duration and maximum thickness. Water residence time is analyzed for the northern and
southern parts of the lagoon as well as the total lagoon area.
The analysis was done by combining historical (1975-2005) and future scenario projection (2006-2100) periods. That
is, two periods/scenarios were assessed: RCP4.5 and RCP8.5, both ranging from 1975 to 2100. This approach
facilitated a comprehensive assessment of the above-mentioned environmental parameters, enhancing insight into
trends and potential variations over time.
In our analysis, we examined the variability of different model runs and the presence of trends and their statistical
significance, as indicated by the *p*-values, across various environmental parameters under different climate models
and scenarios. For this, we applied the Mann-Kendall trend analysis (Hussain and Mahmud, 2019). A *p*-value less
than 0.05 was considered statistically significant. The rate of change was quantified using the Theil-Sen estimator
(Hussain and Mahmud, 2019). The trend analysis was conducted on model outputs, which were aggregated as yearly
means or, in the case of precipitation, as yearly sum.
The timing of spring peak flows was estimated by computing a 3-day moving average of the discharge of Nemunas
River to the delta region. The day of the maximum value during the typical spring flood window occurrence (from the
start of February to the end of April) was noted for each year. The trend was calculated using the same Mann-Kendall
trend analysis approach as described above, using the Julian day of peak flow for each year in the simulation period.
We analyzed the average annual export of Total Nitrogen (TN) and Total Phosphorus (TP) from the Nemunas River
into the Curonian Lagoon. We assessed the trends using the Mann-Kendall test and the 10-year moving averages.
These outputs were compared to the Nutrient Ceiling for the Nemunas River requirements outlined in the HELCOM
Baltic Sea Action Plan (HELCOM, 2021), which are 29338 t year$^{-1}$ for TN and 914 t year$^{-1}$ for TP. We further
evaluated the feasibility of meeting these targets under the conditions of different scenarios and climate models.
The variability between the models, i.e., uncertainty, was assessed by computing the standard deviation and coefficient
of variation using annual values over the entire investigation period (1975-2100). These metrics were based on the
yearly average values of modeled parameters (or sum in case of precipitation, ice season duration, and saltwater
intrusions) for each of the four simulation results using meteorological forcing data from different climate models.

## 2.4.2 The possible impact on fish recruitment success

To evaluate the extent and possible impact of climate change on fish recruitment success, the analysis of the burbot
spawning period was carried out. Burbot requires very cold temperatures (<2°C) for spawning and egg development
(Harrison et al., 2016; Ashton et al., 2019). Within the Curonian Lagoon, it moves to spawning habitats in the Nemunas
River delta. Spawning is most intense at the lowest water temperature (close to 0°C) during December-February,
usually under ice. The duration of the cold period in the projected time series of temperatures suitable for burbot
spawning was calculated by summing days when temperature was below 1.5°C for a given year (days in December
were added to the next year). The R package *changepoint* (Killick and Eckley, 2014, Killick et al., 2022) was used to
estimate the number and locations of change points in a time series of cold period duration. The changes in mean and
variance at a single point were estimated using the *cpt.meanvar* function, employing the AMOC method. The semi-
automatic Pruned Exact Linear Time (PELT) algorithm was employed for the estimation of multiple change point
locations, and parameter estimates within segments (time periods). The number of change points was set to five using
the parameter Q.

## 3 Results

### 3.1 Ensemble dynamics

#### 3.1.1 Water flows

There is a noticeable variability among the climate models in terms of the projected mean yearly water fluxes through the predefined lagoon's cross-sections (Fig. 3). Despite this variability, a consistent pattern emerges across all models, with water outflow from the lagoon towards the sea being a prominent feature in every cross-section examined. Each model captures unique hydrodynamic behaviors at different cross-sections of the lagoon. Still, all indicate that the North of Nemunas and Klaipėda Strait cross-sections generally experience higher water fluxes.

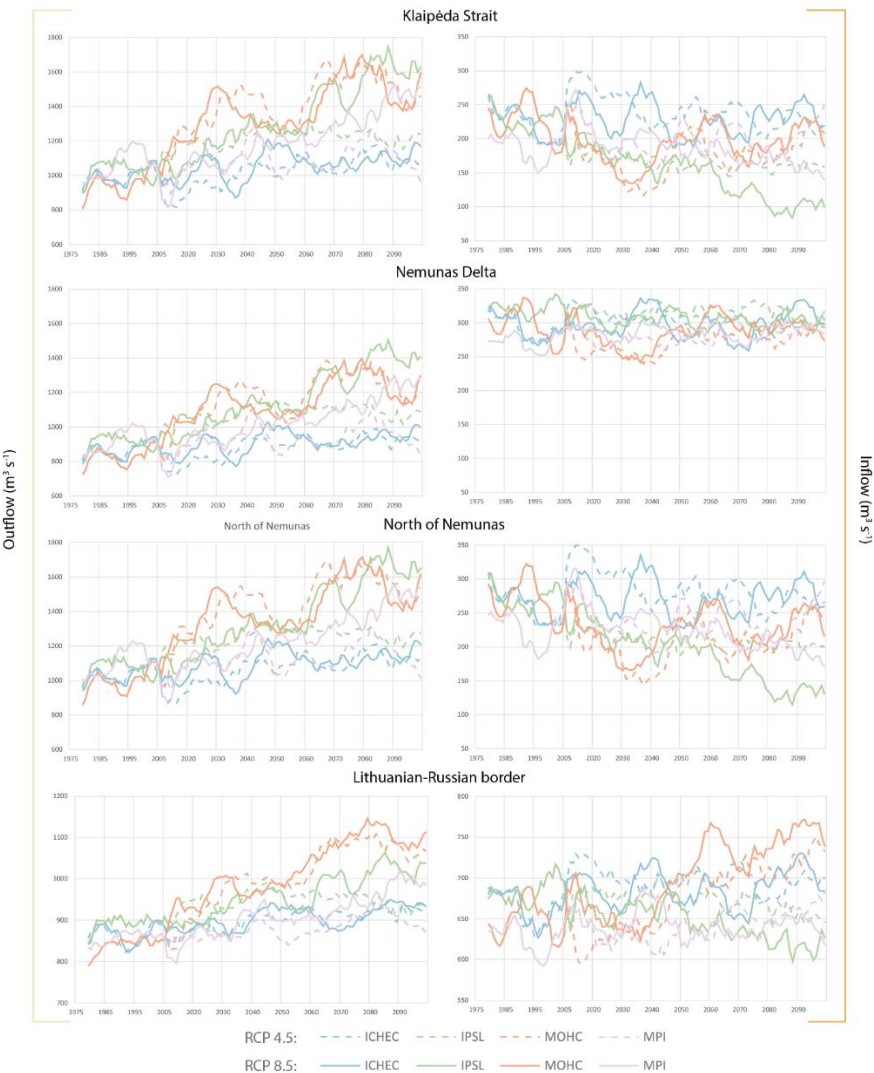

**Figure 3. 10-year moving average graphs of outflowing (left column) and inflowing (right column) water fluxes (in m³ s⁻¹) across four cross-sections within the Curonian Lagoon. Note the adjusted y-axis ranges for fluxes passing through the Lithuanian-Russian border.**

Across all models, the RCP8.5 scenario consistently results in higher mean outflowing water fluxes compared to
RCP4.5, and the MOHC results stand out for consistently projecting the highest mean fluxes in both scenarios,
suggesting a more pronounced increase in water movement through the lagoon compared to its counterparts. Both
scenarios show a higher outflow to the sea discharge with a possible increase of +~300 to +~700 m³ s⁻¹ by the end of
the century. These results could lead to the outflow from the lagoon will reach 37.8-50.4 km³ year⁻¹ which is 24-165
% higher compared to historical outflow.
Regarding the inflowing water fluxes from the Baltic Sea into the Curonian Lagoon, the IPSL model generally predicts
lower fluxes under both scenarios compared to the other models. Inflowing fluxes through the Lithuanian-Russian
border show the least variability in predictions across models, especially under the RCP8.5 scenario, indicating a
consensus on the water flux through this cross-section.
Regarding water residence time (Fig. 4 and Appendix A, Fig. A1 and A2), IPSL tends to predict the shortest water
residence times, suggesting a model inclination towards faster water turnover in the lagoon. In contrast, ICHEC and
MPI, with their higher values, may incorporate factors leading to longer residence times. The shift from RCP4.5 to
RCP8.5 and between different analysis areas does not uniformly affect the models.

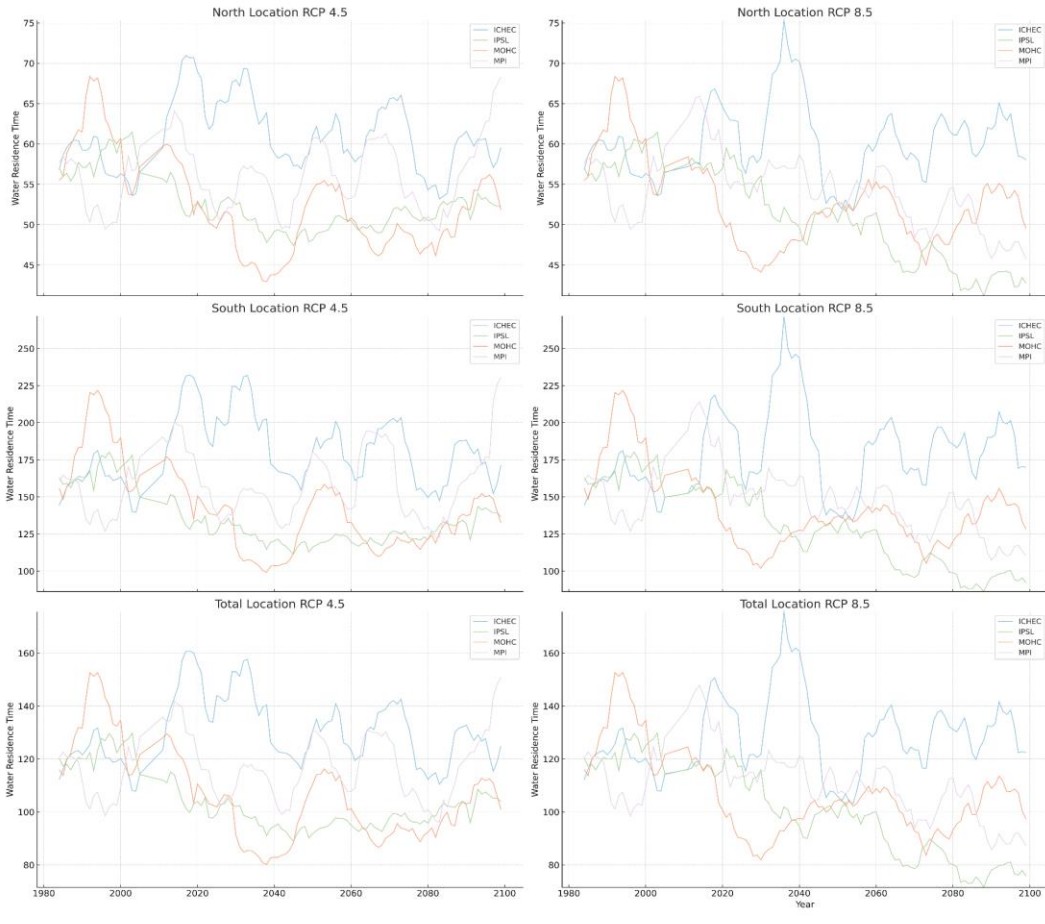


**Figure 4. The timing of the annual water residence times (in days) in the North (upper panels), South (middle panels)**
**parts of the lagoon and the total lagoon area (lower panels) for both RCPs.**

 **3.1.2 Timing of peak flows**

The high discharge of the Nemunas River and subsequent flooding of the delta region is a nearly annual event which
occurs in late winter - spring season and is referred to as "spring flood" in Lithuania. We use the same term in this
study and consider the historic period of high river flows to be from 1st of February to 30th of April. The timing of
spring floods in the Nemunas River delta was previously reported to shift to earlier days due to climate change
(Čerkasova et al., 2021). Further statistical analysis of the projected flows shows that overall there is a statistically
significant relationship between the independent variable 'Year' and the Julian day of occurrence of peak flows in the
Nemunas River for both RCPs when analyzing the entire period (Fig. 5).

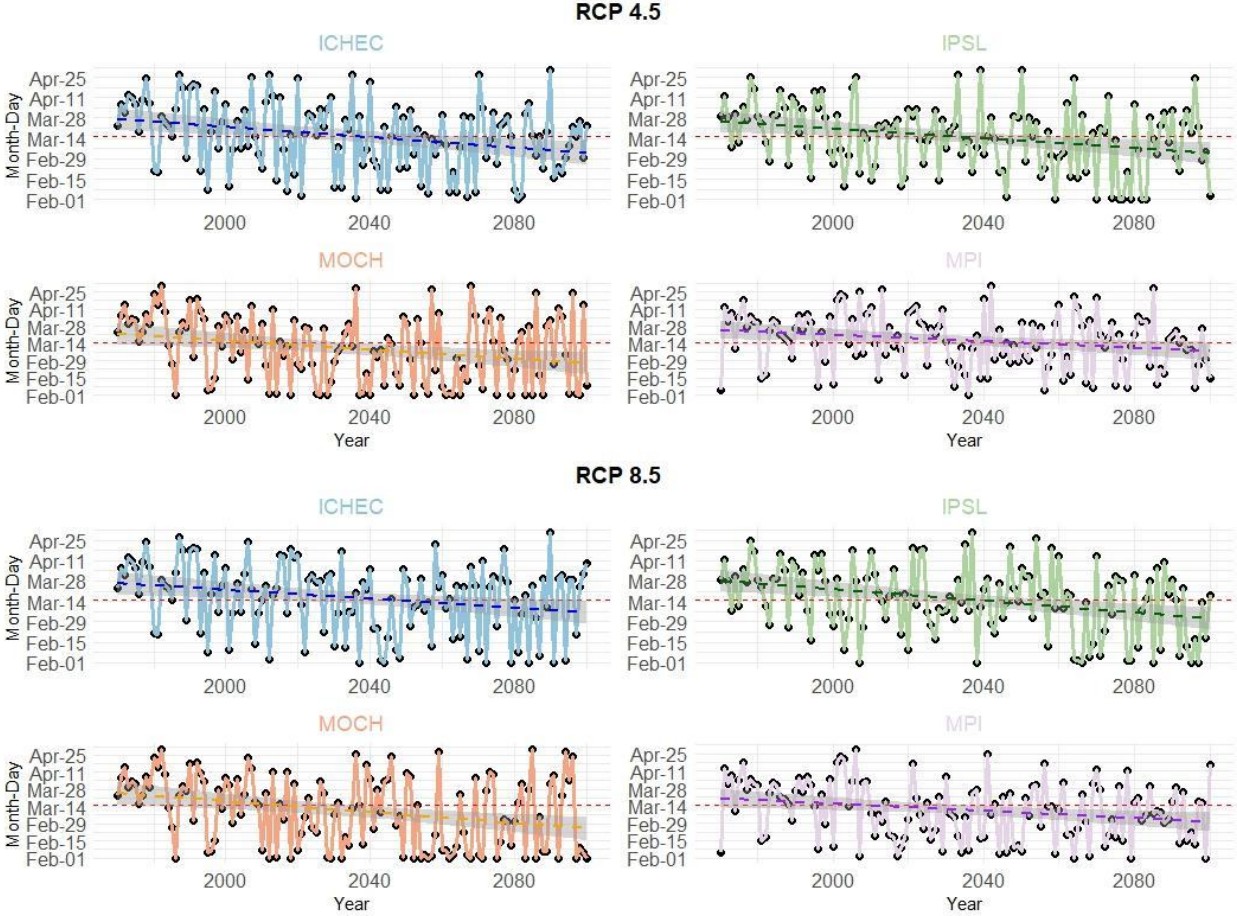


**Figure 5. The timing of occurrence of the average 3-day maximum flow-rate in the spring (between the 1st of February to the 30th of April) of the Nemunas River to the delta region with a trendline for each model. The horizontal red line depicts the period's middle date: March 15th.**

The graphs show that the projected timing of spring high flows is expected to advance under all climate change
scenarios, meaning that the maximum flows are expected to occur earlier in the year. The magnitude of the advance
is greater for the higher emissions scenario (RCP8.5). Both IPSL and MOCH project a higher magnitude of change,
judging by the steepness of the slope, whereas ICHEC and MPI project a moderate rate of change. This could have
several impacts, such as disrupting fish spawning cycles and increasing the risk of flooding.

 **3.1.3 Nutrient loads**

The projections suggest varying levels of variability and trends in the TN and TP loads from the Nemunas River across
different RCPs and models (Fig. 6). The RCP8.5 generally projects higher TN and TP loads compared to RCP4.5 for
all models. This suggests that more extreme climate change scenarios lead to higher nutrient loads in the study region.

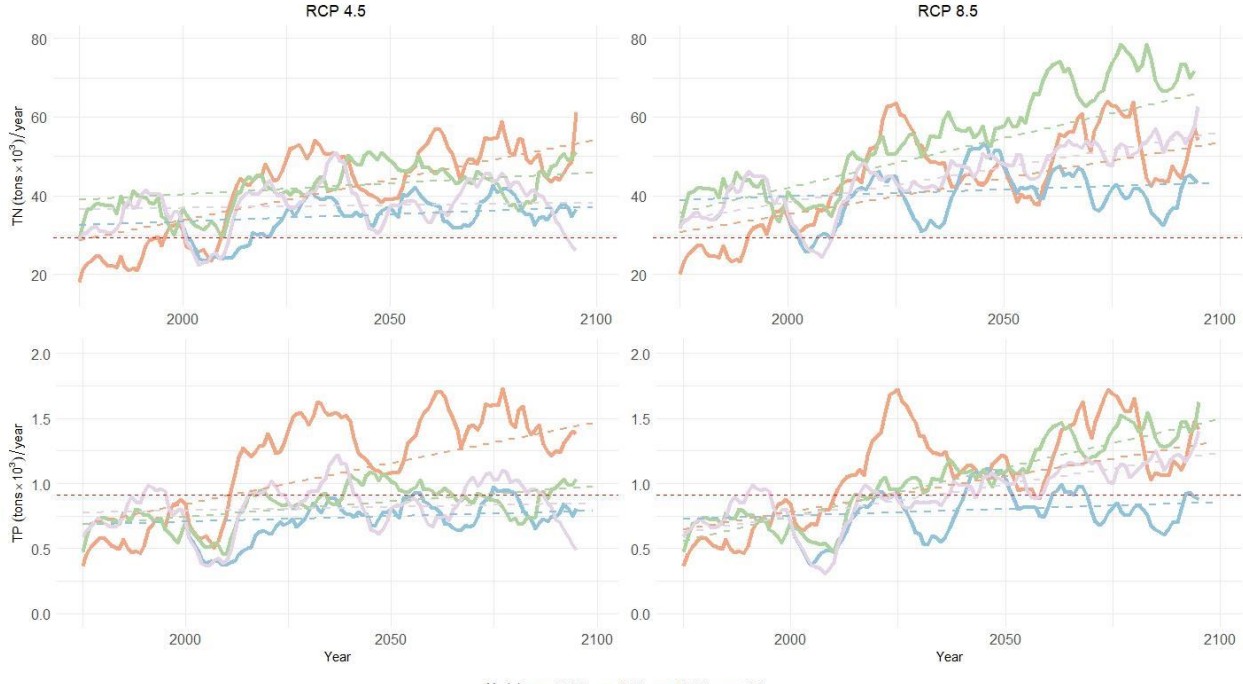

**Figure 6. The projected 10-year moving average of the annual mean TN and TP loads from the Nemunas River to the lagoon**
**with a trendline for each model. The horizontal red line depicts the Revised Nutrient Input Ceiling for the Nemunas River**
**defined by the BSAP update (HELCOM, 2021).**
The projected TN loads are expected to remain above the Revised Nutrient Input Ceiling under all four climate models
and both climate change scenarios (RCP4.5 and RCP8.5) throughout the entire simulation period (shown as a red line
in Fig. 6). Overall, the graph suggests that even under the stabilization scenario (RCP4.5), TN loads from the Nemunas
River are expected to remain above the BSAP (Baltic Sea Action Plan) targets. Total P loads can fall below the
maximum target during several brief periods, but the timing of this will depend on the actual climate scenario that
unfolds. There is substantial variability between the models, which indicates a high level of uncertainty in the
projections. Notably, under the condition of the MOCH model, the TP projections elevate mid-century and further
stabilize at high loads by the end of the modeled period. The IPSL under the RCP8.5 is projecting higher loads,
whereas ICHEC and MPI display a moderate increase (Fig. 6). The lowest average annual nutrient load is projected
under the ICHEC for both RCPs.
**3.1.4 Saltwater intrusions**
The data of the number of days of saltwater intrusion events, i.e., when salinity in Juodkrantė exceeds the 2 g kg$^{-1}$
threshold, shows yearly variations across different models (Fig. 7). All models exhibit considerable year-to-year
variability in the number of saltwater intrusion days, highlighting the complex interplay of climate variability and
local hydrological processes affecting the intrusions. Both ICHEC and MPI often show higher numbers of saltwater
intrusion days compared to IPSL and MOHC. When comparing the RCP4.5 scenario with RCP8.5, the models yield
varying results – ICHEC and IPSL show a slight decrease in intrusion days, MOHC slightly increases, and MPI shows
a moderate decrease.


**Figure 7. 10-year moving average of the number of days of saltwater intrusions (salinity exceeding 2 g kg-1 threshold) reaching Juodkrantė. Underlying time series denote annual saltwater intrusions.**

## 3.1.5 Water temperature

The annual mean water temperature within the lagoon and adjacent coastal areas is depicted in Fig. 8. Under the severe
RCP8.5 scenario, hydrodynamic model simulations predict a noticeable increase in both mean water temperatures and
their variability compared to the RCP4.5 scenario, indicating higher temperatures with greater uncertainty ahead. The
IPSL model consistently projects slightly warmer temperatures across scenarios, while the MOHC model shows the
largest jump in variability, suggesting that it predicts greater uncertainty under RCP8.5. Despite model variations, the
trend towards warmer and more uncertain climate conditions is universally acknowledged.

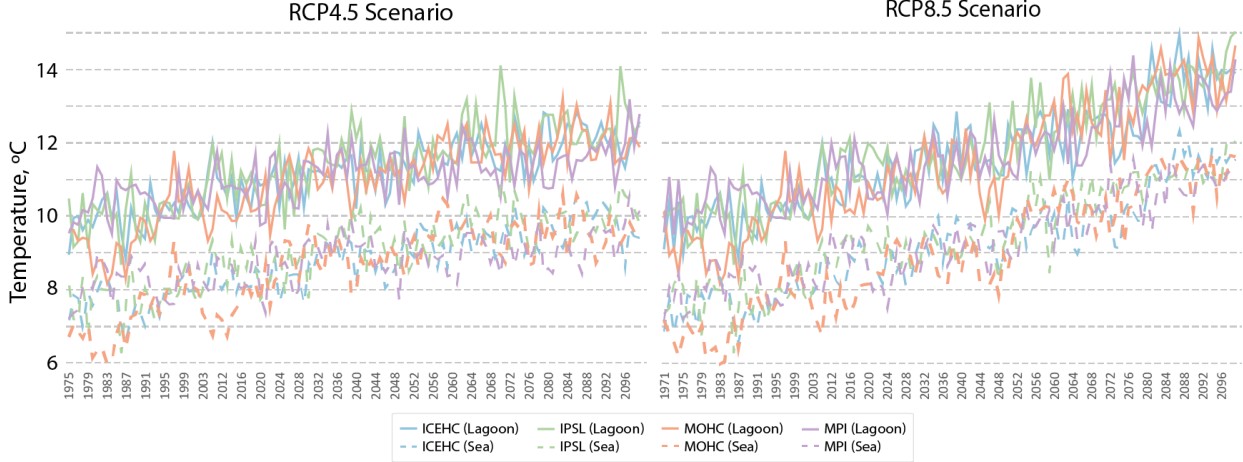


**Figure 8. 10-year moving average of annual mean water temperature in the Curonian Lagoon and southeastern coastal**

**area of the Baltic Sea.**
**3.1.6 Ice thickness**
The comparative analysis of climate model projections for maximum ice thickness and ice season duration (Fig. 9)
highlights the diverse outcomes projected by different model simulations over time and through various scenarios. All
models indicate a shortening of the ice season and thinning of the ice. Notably, the MOHC model often showed lower
thicknesses and shorter ice season duration compared to other models with a distinctive sinusoidal pattern. In contrast,
ICHEC indicates a more gradual decline in ice season duration, whereas IPSL and MPI exhibit a greater variability
over the years. Regarding maximum ice thickness, ICHEC and MPI show higher year-to-year variability.

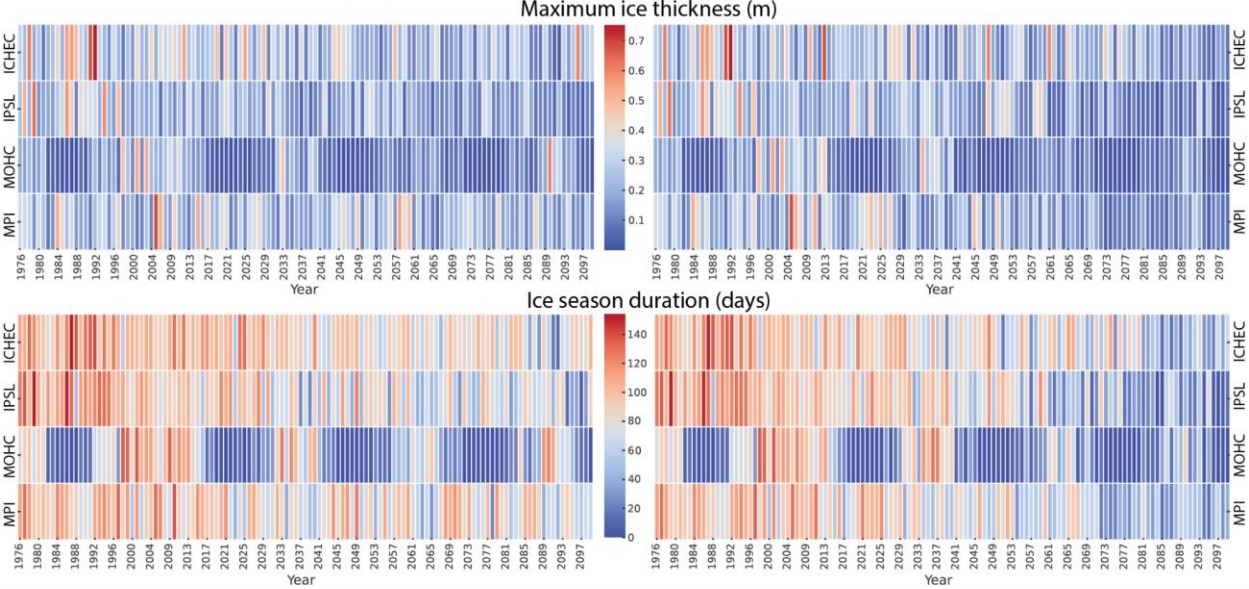


**Figure 9. Heatmaps of maximum ice thickness and ice season duration in the Curonian Lagoon. RCP4.5 (left) and RCP8.5**
**(right).**

## 3.2 Trend analysis

Figure 10 provides a comprehensive overview of trends, accompanied by their statistical significance (*p*-values), and the rate of change (Theil-Sen estimator) for various environmental parameters under different climate scenarios. The results revealed that numerous parameters exhibited significant trends over time. Notably, air temperature and precipitation consistently show significant increasing trends in all scenarios. Although the rate of change varies among the climate models, precipitation exhibits a more pronounced increase compared to air temperature. Water temperature and water level also consistently exhibit increasing trends in all scenarios.

| Parameter | | Presence of trend and its significance (p-value) — Historical + RCP 4.5 | | | | | Historical + RCP 8.5 | | | | | The rate of change (Theil-Sen estimator) — Historical + RCP 4.5 | | | | | Historical + RCP 8.5 | | | | |
|---|---|---|---|---|---|---|---|---|---|---|---|---|---|---|---|---|---|---|---|---|---|
| | | ICHEC | IPSL | MOHC | MPI | Mean | ICHEC | IPSL | MOHC | MPI | Mean | ICHEC | IPSL | MOHC | MPI | Mean | ICHEC | IPSL | MOHC | MPI | Mean |
| Air temperature (°C) | | <0.01 | <0.01 | <0.01 | <0.01 | <0.01 | <0.01 | <0.01 | <0.01 | <0.01 | 0.02 | 0.03 | 0.03 | 0.03 | 0.02 | 0.03 | 0.04 | 0.05 | 0.05 | 0.03 | 0.04 |
| Precipitation (mm year⁻¹) | | <0.01 | <0.01 | <0.01 | 0.02 | <0.01 | <0.01 | <0.01 | <0.01 | <0.01 | <0.01 | 1.00 | 1.64 | 1.48 | 0.68 | 1.21 | 0.90 | 3.21 | 2.14 | 2.07 | 2.08 |
| Water outflow from the lagoon (m³ s⁻¹) | Klaipėda Strait | 0.04 | <0.01 | <0.01 | 0.35 | <0.01 | 0.01 | <0.01 | <0.01 | <0.01 | <0.01 | 1.19 | 2.34 | 6.41 | 0.59 | 2.72 | 1.57 | 6.68 | 6.08 | 4.32 | 4.79 |
| | North of Nemunas | 0.02 | <0.01 | <0.01 | 0.37 | <0.01 | 0.01 | <0.01 | <0.01 | <0.01 | <0.01 | 1.20 | 2.31 | 6.21 | 0.57 | 2.67 | 1.56 | 6.50 | 5.91 | 4.20 | 4.68 |
| | Nemunas Delta | 0.03 | <0.01 | <0.01 | 0.46 | <0.01 | 0.01 | <0.01 | <0.01 | <0.01 | <0.01 | 0.85 | 2.13 | 4.83 | 0.41 | 2.11 | 1.26 | 5.58 | 4.60 | 3.44 | 3.79 |
| | LT-RU border | <0.01 | <0.01 | <0.01 | <0.01 | <0.01 | <0.01 | <0.01 | <0.01 | <0.01 | <0.01 | 0.56 | 0.62 | 2.82 | 0.46 | 1.13 | 0.69 | 1.73 | 2.93 | 1.34 | 1.67 |
| Water inflow from the sea (m³ s⁻¹) | Klaipėda Strait | 0.47 | <0.01 | 0.02 | 0.19 | 0.01 | 0.57 | <0.01 | 0.32 | <0.01 | <0.01 | -0.16 | -0.75 | -0.41 | 0.22 | -0.25 | -0.11 | -1.39 | -0.27 | -0.46 | -0.54 |
| | North of Nemunas | 0.50 | <0.01 | 0.01 | 0.25 | 0.01 | 0.66 | <0.01 | 0.01 | <0.01 | <0.01 | -0.16 | -0.77 | -0.57 | 0.21 | -0.28 | -0.09 | -1.52 | -0.42 | -0.57 | -0.64 |
| | Nemunas Delta | 0.38 | 0.20 | 0.74 | 0.02 | 0.98 | 0.97 | 0.07 | 0.83 | 0.33 | 0.70 | -0.10 | -0.15 | -0.05 | 0.21 | 0.00 | 0.00 | -0.21 | -0.04 | 0.11 | -0.03 |
| | LT-RU border | 0.38 | <0.01 | <0.01 | 0.03 | <0.01 | 0.37 | <0.01 | <0.01 | 1.00 | <0.05 | 0.11 | -0.25 | 0.88 | 0.35 | 0.28 | 0.13 | -0.57 | 1.11 | 0.00 | 0.05 |
| Max spring flow (Julian day) | | <0.01 | <0.01 | <0.01 | <0.01 | <0.01 | <0.01 | <0.01 | <0.01 | <0.01 | <0.01 | -0.18 | -0.17 | -0.20 | -0.15 | -0.28 | -0.14 | -0.19 | -0.23 | -0.15 | -0.28 |
| Nutrients (T year⁻¹) | Total Nitrogen | 0.01 | <0.01 | <0.01 | 0.09 | <0.01 | 0.04 | <0.01 | <0.01 | <0.01 | <0.01 | 65.16 | 121.62 | 287.07 | 49.20 | 57.11 | 65.94 | 388.51 | 251.99 | 217.60 | 88.94 |
| | Total Phosphorus | 0.03 | <0.01 | <0.01 | 0.14 | <0.01 | 0.07 | <0.01 | <0.01 | <0.01 | <0.01 | 1.69 | 3.23 | 9.41 | 0.51 | 3.68 | 1.66 | 9.04 | 7.48 | 5.72 | 6.23 |
| Nemunas River discharge (m³ s⁻¹) | | 0.01 | <0.01 | <0.01 | 0.47 | <0.01 | 0.01 | <0.01 | <0.01 | <0.01 | <0.01 | 1.09 | 2.30 | 4.80 | 0.51 | 2.11 | 1.37 | 5.71 | 4.39 | 3.59 | 3.73 |
| Water temperature (°C) | SE Baltic Sea | <0.01 | <0.01 | <0.01 | <0.01 | <0.01 | <0.01 | <0.01 | <0.01 | <0.01 | 0.01 | 0.02 | 0.02 | 0.03 | 0.01 | 0.02 | 0.03 | 0.03 | 0.04 | 0.03 | 0.03 |
| | Curonian Lagoon | <0.01 | <0.01 | <0.01 | <0.01 | <0.01 | <0.01 | <0.01 | <0.01 | <0.01 | 0.01 | 0.02 | 0.02 | 0.03 | 0.01 | 0.02 | 0.04 | 0.04 | 0.04 | 0.03 | 0.04 |
| Burbot spawning period (days, t<1.5°C) | | <0.01 | <0.01 | <0,01 | <0.01 | <0.01 | <0.01 | <0.01 | <0.01 | <0.01 | <0.01 | -0.51 | -0.49 | -0.27 | -0.23 | -0.40 | -0.60 | -0.56 | -0.32 | -0.41 | -0.53 |
| Water level (cm) | SE Baltic Sea | <0.01 | <0.01 | <0.01 | <0.01 | <0.01 | <0.01 | <0.01 | <0.01 | <0.01 | <0.01 | 0.20 | 0.09 | 0.98 | 0.16 | 0.36 | 0.27 | 0.15 | 1.07 | 0.21 | 0.42 |
| | Curonian Lagoon | <0.01 | <0.01 | <0.01 | <0.01 | <0.01 | <0.01 | <0.01 | <0.01 | <0.01 | 0.03 | 0.21 | 0.12 | 1.00 | 0.16 | 0.37 | 0.28 | 0.24 | 1.07 | 0.25 | 0.46 |
| Ice | Season duration (days) | <0.01 | <0.01 | 0.02 | <0.01 | <0.01 | <0.01 | <0.01 | <0.01 | <0.01 | <0.01 | -0.35 | -0.53 | -0.17 | -0.25 | -0.29 | -0.59 | -0.87 | -0.28 | -0.60 | -0.64 |
| | Max thickness (cm) | <0.01 | <0.01 | 0.17 | 0.09 | <0.01 | <0.01 | <0.01 | <0.01 | <0.01 | <0.01 | -0.11 | -0.13 | -0.03 | -0.04 | -0.09 | -0.12 | -0.17 | -0.06 | -0.14 | -0.14 |
| Salinity in Juodkrantė >2 g kg⁻¹ (days) | | 0.07 | <0.01 | <0.01 | 0.95 | <0.01 | 0.18 | <0.01 | <0.01 | <0.01 | <0.01 | -0.16 | -0.29 | -0.39 | 0.00 | -0.38 | -0.11 | -0.43 | -0.40 | -0.19 | -0.45 |
| Water residence time (days) | Northern part of the lagoon | 0.87 | 0.03 | <0.05 | 0.40 | 0.18 | 0.64 | <0.01 | 0.01 | <0.01 | <0.01 | 0.01 | -0.06 | -0.08 | 0.03 | -0.02 | 0.02 | -0.16 | -0.06 | -0.09 | -0.08 |
| | Southern part of the lagoon | 0.55 | <0.01 | 0.03 | 0.64 | 0.08 | 0.33 | <0.01 | 0.01 | <0.01 | <0.01 | 0.09 | -0.29 | -0.36 | 0.07 | -0.12 | 0.15 | -0.72 | -0.27 | -0.38 | -0.36 |
| | Total lagoon area | 0.68 | 0.01 | 0.02 | 0.56 | <0.05 | 0.47 | <0.01 | 0.03 | <0.01 | <0.01 | 0.04 | -0.18 | -0.24 | 0.06 | -0.08 | 0.07 | -0.47 | -0.19 | -0.25 | -0.24 |

increasing    decreasing    no trend

**Figure 10. Mann-Kendall trend analysis results. Trends and their significance (p-values assessed at a 0.05 confidence level) with their rate of change (Theil-Sen estimator) of key environmental parameters throughout historical and RCP4.5 and 8.5 scenarios in different geographical locations within the Curonian Lagoon and southeastern (SE) Baltic Sea. Cells are colored based on the direction of the trend.**

The MPI model exhibited the most frequent instances of statistically insignificant trends across the projected parameters. Notably, water inflow/outflow, nutrient discharge, riverine discharge, ice thickness, salinity, and water residence times all failed to meet the $p < 0.05$ significance threshold. Interestingly, the MPI model produced the highest *p*-value (0.02) for precipitation, which is the primary driver of other hydrological and hydrodynamic conditions in the model. It is worth noting that if the threshold for statistical significance were further reduced (e.g., $p < 0.01$), the results for the MPI model could be entirely dismissed. This highlights the importance of carefully considering the chosen significance level when interpreting model outputs.

Theil-Sen slope estimates reveal a consistent pattern of increasing river discharge, nutrient loads, and water outflow across all projections. Conversely, consistent with these rising outflows, negative slopes were observed for inflows

from the sea and salinity. These findings collectively suggest a projected increase in freshwater input to the Curonian

Lagoon, potentially impacting its biological communities.

Figure 10 highlights a critical limitation of analyzing ensemble means alone: it can obscure the heterogeneity present

within individual model projections. This is evident in the water inflow at the LT-RU border, where two models show

statistically insignificant trends, yet the ensemble mean indicates a significant trend. Similarly, the individual model

slopes for IPSL (-0.25) and ICHEC (0.88) portray contrasting projections (decrease vs. increase) compared to the

ensemble mean (0.28) which leans towards an increase. These observations emphasize the importance of considering

the spread of individual model projections and their uncertainties, rather than solely relying on the ensemble mean.

## 3.3 Variability in the projections

Analysis of standard deviations (SD) offers a comprehensive insight into the variations across simulation results using

forcing from different climate models, while coefficients of variation (CV) provide a standardized measure of relative

variability across the assessed environmental parameters (Fig. 11). Air and water temperatures have relatively low SD

values. However, the deviation is more pronounced under the RCP8.5 scenario. Additionally, the SD is higher for air

temperature compared to water temperature. In the case of precipitation, the SD presents more diverse results between

the models, adding to the uncertainty of the modeling results.

| Parameter | | Hist + RCP 4.5 Model | | | | | Hist + RCP 8.5 Model | | | | |
|---|---|---|---|---|---|---|---|---|---|---|---|
| | | ICHEC | IPSL | MOHC | MPI | Mean | ICHEC | IPSL | MOHC | MPI | Mean |
| Air temperature ($^{\circ}$C) | | 1.26 | 1.48 | 1.38 | 0.95 | 1.09 | 1.77 | 1.90 | 1.95 | 1.46 | 1.65 |
| Precipitation (mm year$^{-1}$) | | 90.27 | 129.52 | 119.04 | 98.47 | 70.44 | 105.48 | 170.57 | 131.61 | 127.41 | 92.36 |
| Water outflow from the lagoon (m$^3$ s$^{-1}$) | Klaipėda Strait | 171.84 | 195.43 | 347.72 | 186.72 | 147.45 | 194.88 | 327.40 | 336.50 | 266.75 | 194.74 |
| | North of Nemunas | 165.65 | 189.56 | 338.24 | 180.07 | 143.54 | 187.32 | 318.29 | 326.21 | 258.11 | 189.55 |
| | Nemunas Delta | 140.26 | 167.89 | 276.13 | 151.39 | 119.50 | 156.58 | 267.42 | 257.70 | 216.96 | 155.28 |
| | LT-RU border | 62.39 | 72.35 | 118.73 | 59.73 | 51.40 | 65.05 | 94.00 | 123.68 | 84.09 | 64.49 |
| Water inflow from the sea (m$^3$ s$^{-1}$) | Klaipėda Strait | 53.25 | 51.93 | 69.04 | 54.31 | 30.88 | 60.26 | 66.17 | 59.70 | 55.34 | 32.45 |
| | North of Nemunas | 58.15 | 56.42 | 77.18 | 59.85 | 34.42 | 66.37 | 73.41 | 66.70 | 62.40 | 36.37 |
| | Nemunas Delta | 40.88 | 43.67 | 47.12 | 38.76 | 21.56 | 46.64 | 46.45 | 49.00 | 41.57 | 22.40 |
| | LT-RU border | 52.68 | 54.24 | 75.50 | 52.92 | 29.98 | 55.76 | 64.83 | 72.79 | 56.40 | 29.71 |
| Nemunas River discharge (m$^3$ s$^{-1}$) | | 153.49 | 180.21 | 297.70 | 174.22 | 139.92 | 180.07 | 287.70 | 272.62 | 230.75 | 170.21 |
| Max spring flow (Julian day) | | 23.36 | 21.96 | 26.66 | 21.79 | 22.24 | 23.83 | 23.04 | 28.05 | 22.60 | 22.61 |
| Nutrients (T year$^{-1}$) | Total Nitrogen | 11398.94 | 14316.05 | 21139.06 | 12646.04 | 4292.64 | 14463.22 | 22077.95 | 20908.10 | 15709.44 | 4729.58 |
| | Total Phosphorus | 335.01 | 357.86 | 623.00 | 374.45 | 291.82 | 382.61 | 503.74 | 623.49 | 417.17 | 339.24 |
| Water temperature ($^{\circ}$C) | SE Baltic Sea | 0.88 | 0.97 | 1.12 | 0.67 | 0.78 | 1.32 | 1.31 | 1.55 | 1.11 | 1.24 |
| | Curonian Lagoon | 0.90 | 1.08 | 1.11 | 0.71 | 0.81 | 1.36 | 1.44 | 1.64 | 1.15 | 1.30 |
| Burbot spawning period (days, t<1.5$^{\circ}$C) | | 28.56 | 28.3 | 25.22 | 22.1 | 18.7 | 31.67 | 29.84 | 26.07 | 24.79 | 22.09 |
| Water level (cm) | SE Baltic Sea | 0.09 | 0.07 | 0.34 | 0.08 | 0.13 | 0.11 | 0.08 | 0.38 | 0.09 | 0.15 |
| | Curonian Lagoon | 0.09 | 0.08 | 0.35 | 0.08 | 0.11 | 0.11 | 0.11 | 0.38 | 0.11 | 0.14 |
| Ice | Season duration (days) | 24.28 | 30.29 | 42.67 | 24.45 | 18.84 | 30.83 | 38.01 | 41.97 | 29.80 | 29.37 |
| | Max thickness (m) | 0.14 | 0.12 | 0.12 | 0.12 | 0.06 | 0.15 | 0.13 | 0.12 | 0.13 | 0.07 |
| Salinity in Juodkrantė >2 g kg$^{-1}$ (days) | | 25.95 | 21.19 | 25.76 | 21.55 | 18.72 | 28.13 | 22.60 | 24.96 | 20.66 | 21.20 |
| Water residence time (days) | Northern part of the lagoon | 10.60 | 7.51 | 10.52 | 10.13 | 5.13 | 12.14 | 9.62 | 9.18 | 9.40 | 5.41 |
| | Southern part of the lagoon | 59.86 | 38.26 | 51.89 | 62.65 | 26.38 | 73.21 | 45.45 | 44.74 | 48.49 | 27.99 |
| | Total lagoon area | 32.24 | 22.55 | 30.37 | 29.71 | 14.84 | 37.88 | 27.60 | 26.23 | 27.98 | 16.12 |

Coefficient of variation: <10% | 10-20% | 20-30% | 30-40% | 40-50% | 50-60% | 60-70% | 70-80% | 80-90% | >90%

**Figure 11. Standard deviations of key environmental parameters throughout historical and RCP4.5 and 8.5 scenarios in different geographical locations within the Curonian Lagoon and southeastern (SE) Baltic Sea. Cells are colored based on the coefficient of variation.**

The low *p*-values (Fig. 10) indicate that the trends in earlier maximum spring flows are statistically significant.
Variability in the SD values across models and RCPs (Fig. 11) suggests that there is uncertainty associated with these
projections. The range of Theil-Sen slopes also indicates variability in the rate of decline in the timing of maximum
spring flows across different scenarios. Therefore, while the trends are significant, the variability in the projections
should be considered when interpreting and using these results for decision-making. The SD values for the occurrence
of maximum spring flows range from 21.79 (in the MPI 4.5 scenario) to 28.05 days (in the MOHC 8.5 scenario),
where higher SD values indicate greater variability in the predicted time series data. Both IPSL and MPI models have
lower prediction variability, whereas MOHC and ICHEC display larger variability. The RCP8.5 scenario indicates a
greater degree of change, consistent with previous studies (Idzelytė et al., 2023a, Čerkasova et. al., 2021). For both
RCP4.5 and RCP8.5, the MPI model has the lowest CV (29% and 32%), while the MOHC model has the highest CV
(38% and 40%). Based on these results it can be concluded that the MPI model appears to be less variable compared
to the other models for both RCP4.5 and RCP8.5 scenarios. Conversely, the MOHC model appears to be more variable
compared to the other models for both scenarios.
Analysis of potential future TN and TP loads in the Nemunas River reveals a broad spectrum of possibilities. The
variation is linked to the specific climate model and RCP scenario chosen. However, a consistent trend emerges across
all models and RCPs – an upward trajectory for nutrient loads. Anthropogenic activities are the primary driver of
nutrient loading from land sources. While climate factors, such as increased precipitation and subsequent nutrient
wash-off, might exert a net negative impact on loads, a comprehensive future outlook requires incorporating
anticipated changes in nutrient management practices and land use. This study acknowledges the omission of these
factors, highlighting the need for further analysis to identify the most probable scenario and develop potential
mitigation strategies for nutrient pollution in the Nemunas River.
When examining water dynamics within the lagoon, areas with greater fluctuations in SD are notably found in regions
where water flow is more intense. This pattern is particularly distinguished from the Nemunas Delta going northward
to the Klaipėda Strait. Variability is much higher for water outflow than inflow. The most significant variation between
the models is evident under the RCP4.5 scenario, where simulation results derived using MOHC datasets produce
much higher SD than other models. A similar pattern is also evident for the ice season duration, while SD for saltwater
intrusions in the lagoon is relatively similar between the different models. Water residence time exhibits the same
variability between the models in all analysis sections. Notably, the IPSL model demonstrates a lower SD under the
RCP4.5 scenario, while the ICHEC model exhibits a higher SD under the RCP8.5 scenario.
In almost all instances, except for water level, the SD statistics derived from the model-averaged datasets exhibit lower
values. This suggests a reduction in variability compared to individual models, emphasizing the smoothing effect
achieved through model averaging. The most pronounced disparity in SD among the models is observed in the case
of MOHC, particularly regarding the RCP4.5 scenario.
The differences between climate models become more apparent when considering coefficients of variation. While air
and water temperatures show relatively consistent results with low CV values, parameters like salinity and ice-related
variables display higher CV values, highlighting greater variability and uncertainty among the climate models. Among
the parameters indicating water flow dynamics in different areas of the Curonian Lagoon, again a clear disparity of
the MOHC model can be seen. This indicates the model's distinct response and emphasizes the need for careful
consideration when employing the data of this climate model in hydrodynamic and hydrological simulations.

**3.4 Changepoint analysis of burbot spawning period time series**

The single changepoint analyses of major shifts in mean and variance in time series of the duration of the cold season
suitable for burbot spawning occur from 2013 to 2029 according to modeling results of RCP4.5 (Appendix B Fig.
B1). The mean value of the time segment involving historical and recent past varies from 47 (MOHC RCP4.5) to 72
days (ICHEC RCP4.5). In the next period, it becomes shorter by 66% according to MOHC and IPSL models and by
36 and 51% according to MPI and ICHEC models, respectively, taking no longer than one month. In three of the four
RCP8.5 scenario models the single changepoint could only be detected at the end of the time series, after 2040-2060
when the cold period duration is reduced to 6 to 9 days. An exception was generated by ICHEC RCP8.5 model results,
indicating a changepoint in the historical past, showing that the duration of the cold period already decreased by 60%
in 1992 (Appendix B Fig. B1). Somewhat surprisingly, no changepoints in terms of variance are detected in the IPSL
and ICHEC time series. Change in variance was detected in the IPSL time series in 1995 and in the MOHC time series
in 2003 and 2013 (Appendix B Fig. B2), so both occurred within the historical period. Both model results indicate a
2 to 3 times higher variance of the cold period duration in the historical period than post changepoint period (Appendix
B Fig. B2).

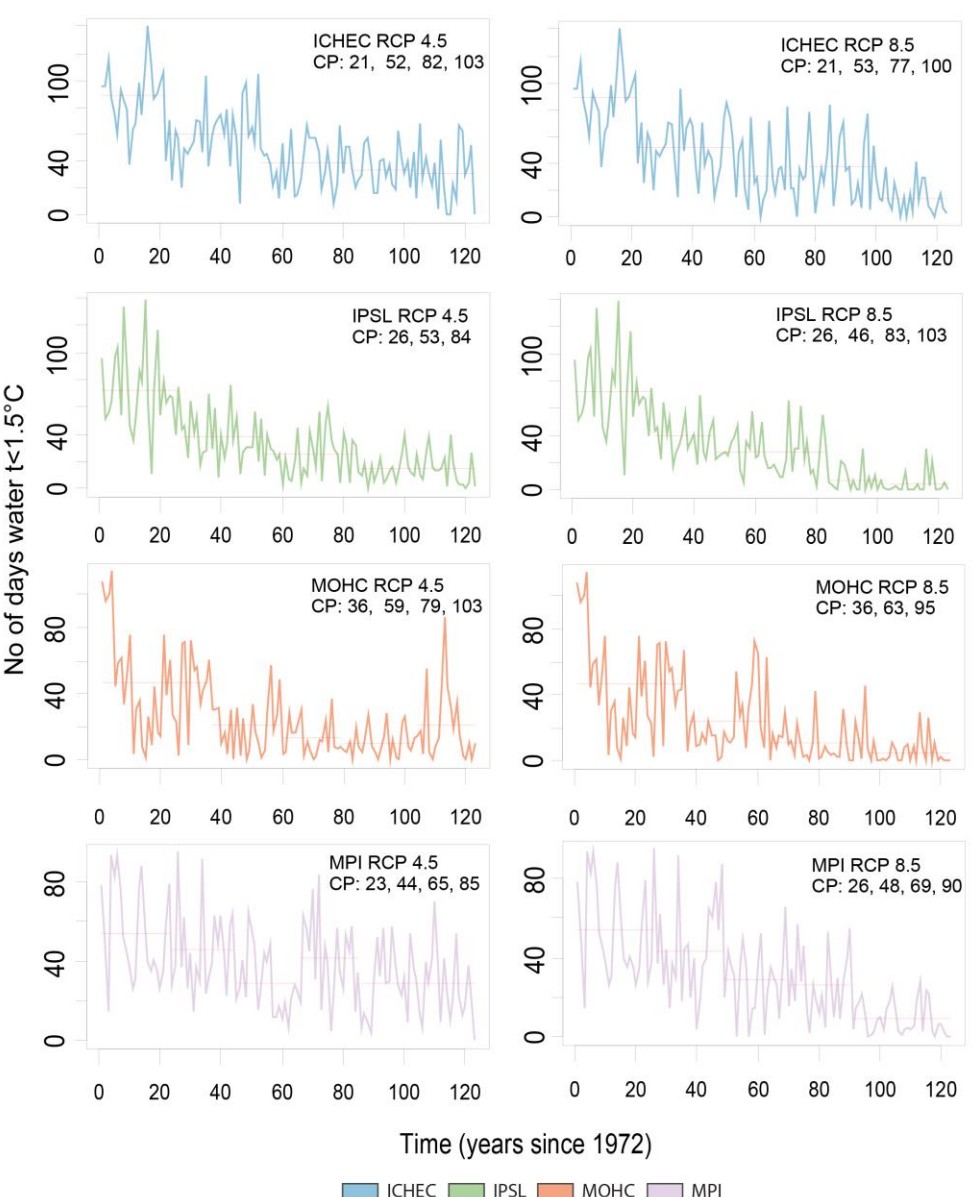


**Figure 12. Changepoint (CP) detection in the modeled time series of burbot spawning period t<1.5ºC duration (Vente area). CP refers to changepoints indicated in a number of years since 1972.**

Multiple changepoint detection analyses indicated three to four changepoints in the modeled time series of cold period duration (Fig. 12). The significant decrease in a mean number of <1.5 ºC days occurred in the 1990s according to MPI, IPSL, and ICHEC models, and the change was particularly obvious in the results of IPSL and ICHEC models 46-47% and 33-42% reduction, respectively (Table 3). The cold period duration decreased from three to two months according to the ICHEC RCP4.5 model and even to less than 2 months in ICHEC RCP8.5 in 1992. The next time segment where all modeled time series had a changepoint is close to the present time and near future (Table 3). After this changepoint the cold period is further shrinking. If in the 1990s the MPI model showed only a slight decrease in the number of cold days (15%), after 2021 (MPI RCP4.5) and 2025 (MPI RCP8.5) the reduction is more severe (46%). After the

second changepoint, 46 to 72% of the initial cold period duration is lost according to all model results. According to
three out of four model results, the cold period duration is less than one month after the 2030s.

| | | | | **Change points & Means of periods** | | | | | |
|---|---|---|---|---|---|---|---|---|---|
| | Mean I | **Historic CP** | Mean II | **Present & Near future 2020-2040 CP** | Mean III | **Long-term 2040-2060 CP** | Mean IV | **Long-term >2060 CP** | Mean V |
| **MOHC 4.5** | 47 | 2013 | 20 | 2036 | 13 | 2056 | 10 | 2080 | 21 |
| **MPI 4.5** | 54 | 1994 | 46 | 2021 | 29 | 2042 | 42 | 2062 | 29 |
| **IPSL 4.5** | 72 | 1997 | 38 | 2030 | 25 | - | 25 | 2061 | 14 |
| **ICHEC 4.5** | 89 | 1992 | 60 | 2029 | 38 | 2059 | 34 | 2080 | 30 |
| **MOHC 8.5** | 46 | 2013 | 24 | 2040 | 11 | - | 11 | 2072 | 5 |
| **MPI 8.5** | 54 | 1997 | 43 | 2025 | 29 | 2046 | 26 | 2067 | 9 |
| **IPSL 8.5** | 72 | 1997 | 39 | 2023 | 28 | 2060 | 7 | 2080 | 5 |
| **ICHEC 8.5** | 89 | 1992 | 52 | 2030 | 31 | 2054 | 37 | 2077 | 14 |

**Table 3. Multiple changepoints (CP, years) in modeled time series and mean values of burbot spawning period duration**
**(number of days when the temperature was <1.5 ºC) in subsequent periods (I-V) at the Vente area.**
**4 Discussion**
Variability and uncertainty are not a flaw but a representative aspect of predicting complex systems. Multi-model
ensembles (MMEs) are a vital tool in managing this uncertainty, providing a more robust and reliable basis for
understanding future climate conditions and informing global efforts to mitigate and adapt to climate change. One
common method to analyze MMEs for climate change impact assessment is ensemble averaging, which is often
considered more accurate than any individual model's prediction, smoothing out model-specific biases. We performed
this type of research in our previous study (Idzelytė et al., 2023a), however, investigating the dynamics of each model
separately is important for evaluating the overall variability in impact predictions since relying only on multi-model
averaging can obscure the detailed representation of extreme values and the variability of the parameters under study,
potentially affecting the accuracy of projections (Tegegne et al., 2020).
In our study, 2 RCPs model scenarios of one RCM driven by 4 GCM were analyzed and each model showed
independent variability of the parameters and its trends. In general, the trends are aligned in the same trajectory but
the slope differs: sharper decreases or increases occur in data series based on RCP8.5 forcing. Our study also indicates
that even under climate mitigation scenario RCP4.5 the changes in hydrological processes and temperature regimes
are significant. The combined analysis of standard deviations and coefficients of variation provides valuable insights
into the divergences between climate models in simulating hydrodynamic and hydrological processes.

## 4.1 Riverine inputs and water flows

The discharge of the Nemunas River exhibits a pronounced and statistically significant increasing trend, accompanied by escalating rates of change. The overall water outflow from the lagoon also reveals increasing trends suggesting changes in hydrological patterns, while water inflow varies in significance across scenarios and locations. The significance of the latter is inconsistent and varies between different climate models, with some of them (depending on the cross-sections) displaying no significant trends, and others indicating a decrease in water inflow.

Results from the 10-year moving average imply much higher variability between the models in the long-term period, which reveals the cumulative effect of the uncertainties and complexity of the system. Our study results differ greatly compared to Jakimavičius et al. (2018) study based on the IPCC (2013) climate models without downscaling (GFDL-CM3, HadGEM2-ES, NorESM1-M). Jakimavičius et al. (2018) study applied the HBV hydrological model and used statistical methods to calculate the Baltic Sea parameters. With these techniques the main following projected outputs were generated: 1) the Nemunas outflow decrease from 22.1 to 15.9 km³ with RCP8.5 scenario; 2) decreasing trend of the outflow to the sea will induce only 0.7% from the reference value; 3) and significant inflow increase to the lagoon due to sea level rise was calculated up to 61.3% higher compared to the reference period (Jakimavičius et al., 2018).

Our study results are in line with Plunge et al. (2022) where the SWAT model with 7 regional climate models was applied to test RCP4.5 and RCP8.5 scenarios. Plunge et al. (2022) study projected the increase of the Nemunas River discharge by 9.7% for RCP4.5 and by 35.4% for the RCP8.5 scenario by the end of the century. The divergent results from various studies show the necessity to evaluate climate change scenarios with care. The use of the regional-bias corrected data has a minor variation in the near future; however, the long-term projections are still uncertain. The trend analysis showed that the MOHC model projected the highest riverine input, as a result, most of the other parameters had more distinguished results compared to other models.

The associated trends in water residence time (WRT) in different parts of the lagoon are diverse, having varying levels of significance and rates of changes. However, most of the RCP4.5 models did not show significant trends except the mean trend for this scenario, while with RCP8.5 models prevailing trends of decreasing water residence time can be observed. The decreasing trends can be explained by the higher Nemunas discharges and the increased outflow from the lagoon to the sea. Moreover, the timing of the maximum spring flood shifting to earlier days in the year could have important implications for the lagoon flushing rate in spring, e.g., the absence of ice jam could profoundly reduce the likelihood of the sudden water level rise and extreme flood event risk. Earlier spring floods and the tendency of shorter WRT in the lagoon could have important implications for biogeochemical cycles, nutrient regimes, and associated phytoplankton primary production peaks and overall nutrient retention capacity. The projections show that the timing of spring high flows are moved to the boundary of the analyzed period (February 1[st]), which indicates that the peak flow rate might occur even earlier in the year. Although not analyzed in this paper, a follow-up study will explore these projections using more appropriate methods for detecting trends in flood timing, i.e. using the circular statistics approaches (Blöschl et al., 2017).

**4.2 Saltwater intrusion into the freshwater system**

The variability of water inflow from the Baltic Sea into the lagoon impacts saltwater intrusions in the northern part of the lagoon and has significant effects in the area, extending to around Juodkrantė, which is situated approximately 20 kilometers southward of the Klaipėda Strait (sea inlet). The duration of saltwater intrusions in this specific area exhibits varying trends and rates of change, with certain scenarios displaying significant decreases in the number of days per year when salinity exceeds 2 g kg$^{-1}$, while others – no significant changes. Analysis of single-model saltwater intrusions showed huge variability between the years, particularly it can be visible in ICHEC and MPI model projections. The large variabilities of the projected future salinity were discussed in other studies as well (Meier et al., 2022a, 2022b), claiming that the considerable uncertainties in all salinity drivers together with the different responses to these drivers cause the variability in the salinity projections. In our study, ICHEC and MPI models for RCP4.5 and ICHEC for RCP8.5 showed no trends suggesting that it is very difficult to project the changes in the future. Worth noting that single model projections of the saline water inflows from the North Sea to the Baltic Sea that can influence the saline water intrusions to the Curonian Lagoon were not analyzed. However, given the significant increase in river discharge is anticipated, the saltwater intrusion into the freshwater system is not likely.

**4.3 Water temperature and ice regime in the Curonian Lagoon**

All models showed a significantly increasing trend for the water temperatures with the highest rate of change for the MOHC model and the lowest change for the MPI model. The analysis of the SD values strongly suggests that water temperature is the most certain parameter and all models agree with the rise of water temperature. In general, all of the Baltic Sea displays the same trends under RCP4.5 and 8.5 projections: the water temperature will increase, and the sea-ice cover extent will decrease (Meier et al., 2022b). The impact of the increased water temperatures will be mostly visible during winter periods and crucial for the coldwater species. We did not analyze the possible upwelling and marine heatwave events that are important for the summer period and can have a significant influence on the ecological status of the lagoon and southern Baltic Sea coasts, which leaves opportunity for future research directions. Ice-related parameter results suggest a consistent and significant decline in ice season duration and maximum ice thickness across multiple climate models and scenarios. Results are in line with Jakimavičius et al. (2020) study accomplished with statistical methods using MPI, MOHC, and ICHEC model inputs for the Curonian Lagoon, where the ice duration was projected to last 35–45 days for RCP4.5 and 3–34 for RCP8.5 with an expected decline of the ice thickness up to 0–13 cm in the long term analysis. In our study, the highest rates of change were expressed by the IPSL model, which was not included in the previous study. Nevertheless, both studies agreed that in the future the ice-covered season will be shorter or even absent (RCP8.5). Decreasing ice cover will affect WRTs (Idzelytė et al., 2023a, 2020) and will have consequences for the lagoon ecosystem.

**4.4 Implications for nutrient load management**

One of the greatest concerns of the environmental managers is the projection of the river nutrient loads into the Curonian Lagoon, which heavily affects eutrophication (Vybernaite-Lubiene et al., 2018, Stakėnienė et al., 2023).

This task also relates to the international commitment to reduce nutrient inputs into the Baltic Sea. According to our
model results (ICHEC, IPSL, MPI), the TP threshold could be achieved during several periods with fluctuating
patterns throughout the entire century if RCP4.5 scenario forcing is ensured. However, a severe discrepancy from the
targeted loads of TN is projected by the middle of the century by all models and especially by MOHC, regardless of
the RCP scenario. Despite the limitations of this study (i.e., not taking the possible land use and management change
into account), a worrying trend emerges with the increasing risk that with current regulations Lithuania will unlikely
meet the nutrient input ceilings defined in the HELCOM Baltic Sea Action Plan during the century.
Some studies demonstrate that future socioeconomic pathways could have a greater effect than climate change on
nutrients inputs to the Baltic Sea (Bartošová et al., 2019). Thus, the policy decisions within the BSAP framework do
not lose their importance, even in the context of climate-induced negative consequences, i.e., climate driven increase
in N loads. Measures designed and implemented can have a significant impact on environmental management
achievements of the threshold targets, especially if combined with emission reduction policy and socio-economic
transition towards more sustainable food and waste systems.
**4.5. Implications for nature protection and conservation**
Our study of climate change prediction uncertainty demands a re-evaluation of past approaches in biodiversity
conservation, highlighting the need for adaptive strategies in this field. Burbot used to be a significant part of the
commercial fish catch in the Curonian Lagoon before the 1990s and still is a very important target for game fishing,
especially under the ice. However, both commercial and recreational catches have fallen, and despite massive
restocking efforts, the stock is not improving. Some authors hypothesized that the main reason for the population
decline is the warming temperatures during the reproduction season (Švagždys, 2002). According to Skersonas et al.
(unpublished report 2019), the fall in catches of burbot in the Curonian Lagoon also coincided with the collapse of the
USSR and uncontrolled fishing at the beginning of the state's creation. According to our analysis, the stock collapse
period in fact corresponds to the presence of temperature changepoint detected in 1994, 1997, and 1992 in different
modeled data sets MPI, IPSL, and ICHEC, respectively. High variance of cold days duration among years during the
historic period was reflected in burbot stocks, the sequence of four to six cold winters was followed by a three to five-
fold increase in burbot catches (Švagždys, 2002). However, along with increasing temperature in the future, the change
between colder and warmer winters is not likely. The absence of ice cover, shift in spring flood timing, and increasing
water temperatures potentially could have implications for fish spawning phenology and spawning habitat quality.
Multiple changepoint detection analysis results showed a significant increase in temperature and shortening of the
cold period starting from the 1990s, indicating the onset of global warming. Assuming 'business as usual' carbon
emission scenario RCP8.5, the next notable decrease in cold period duration, already happened in 2023 (IPSL) or is
happening soon in 2025 (MPI) and 2030 (ICHEC). Thereafter the cold period lasts for as long as one month. Assuming
the emission reduction scenario RCP4.5, i.e., the stabilization of temperature trend, a one-month cold period duration
could be expected to last to the end of the century, according to MPI and ICHEC model results. However, IPSL results,
and especially MOHC results show no improvement even under the climate change mitigation scenario. Loss of ice
and cold isothermal conditions for spawning and egg development would further contribute to a significant decline in
burbot population natural recruitment. The aquaculture-based restocking as a conservation measure rather than a stock
improvement measure would become realistic in the near future.
**5 Conclusions and recommendations**
This study evaluates the output from various climate models to understand hydrological and hydrodynamic changes
in the Nemunas River, Curonian Lagoon, and southeastern Baltic Sea continuum under different climate change
scenarios. It highlights the importance of employing multiple models due to their unique predictions and the inherent
variability and complexity in projecting climate impacts on the analyzed hydrological and hydrodynamic parameters.
The analysis revealed that each model exhibits its own unique variability across all the examined parameters, while
some models show greater degrees of change, others are more stable. Yet, despite these variances, all models
consistently align in their projections and tendencies under the RCP4.5 and RCP8.5 climate change scenarios.
To summarize, the effective management of the Nemunas River – Curonian Lagoon – Baltic Sea continuum in a
changing climate needs a collaborative policy framework. Cross-sectoral working groups, focused on specific
challenges like nutrient management, should combine expertise from agriculture, water resources, and environmental
protection agencies. Engaging multiple stakeholder groups (fishermen, environmental managers, agricultural advisors,
scientists, policymakers, etc.) in designing and implementing climate-resilient practices fosters knowledge sharing
and feedback loops, leading to more effective and socially-accepted solutions. For example, promoting practices that
improve nutrient retention can also reduce runoff and, in turn, reduce the risk or magnitude of floods and protect
biodiversity.
With our study we strongly support development of predictive tools to aid in decision-making, risk assessment and
management. The variability results provide valuable insights to initiate policy updates, enhanced regional cooperation
and coordination, development of climate change indicators and associated revision of national monitoring programs
(e.g., Rose et al., 2023). Our results suggest that much greater efforts to mitigate global climate change are needed to
avoid high costs and difficulties to implement local climate mitigation measures.
**Data availability**
All numerical modeling results are openly available in the Zenodo open data repository
(https://doi.org/10.5281/zenodo.7500744, Idzelytė et al. (2023b)), initially generated in Idzelytė et al. (2023a) and
cited in this manuscript.
**Author contribution**
GU and NC initiated the conceptualization and funding acquisition of the research project. NC, JM and RI performed
the analysis and drafted the paper. RI, NC, JM and JL worked on the visualization of the results. NC, JM, RI prepared

the original manuscript draft with the assistance of JL, GU and AE. All co-authors reviewed the paper and contributed to the scientific interpretation and discussion.

**Competing interests**

The authors declare that they have no conflict of interest.

**Acknowledgements**

This project has received funding from the Research Council of Lithuania (LMTLT), agreement No S-MIP-21-24.

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

**Appendix A: Additional results of the water residence time**

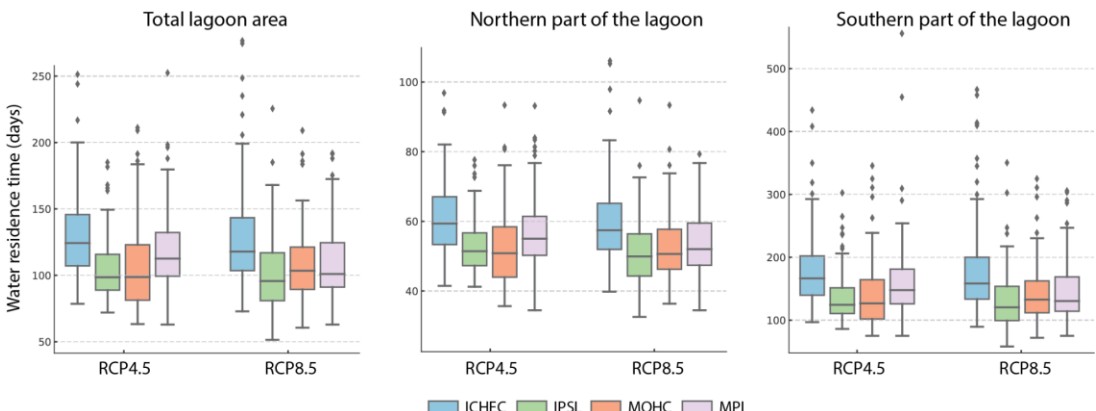


**Figure A1. Annual average water residence time (in days) in the total lagoon area, as well as separately -**
**northern and southern parts of it, under RCP4.5 (left column) and RCP8.5 (right column) scenarios.**

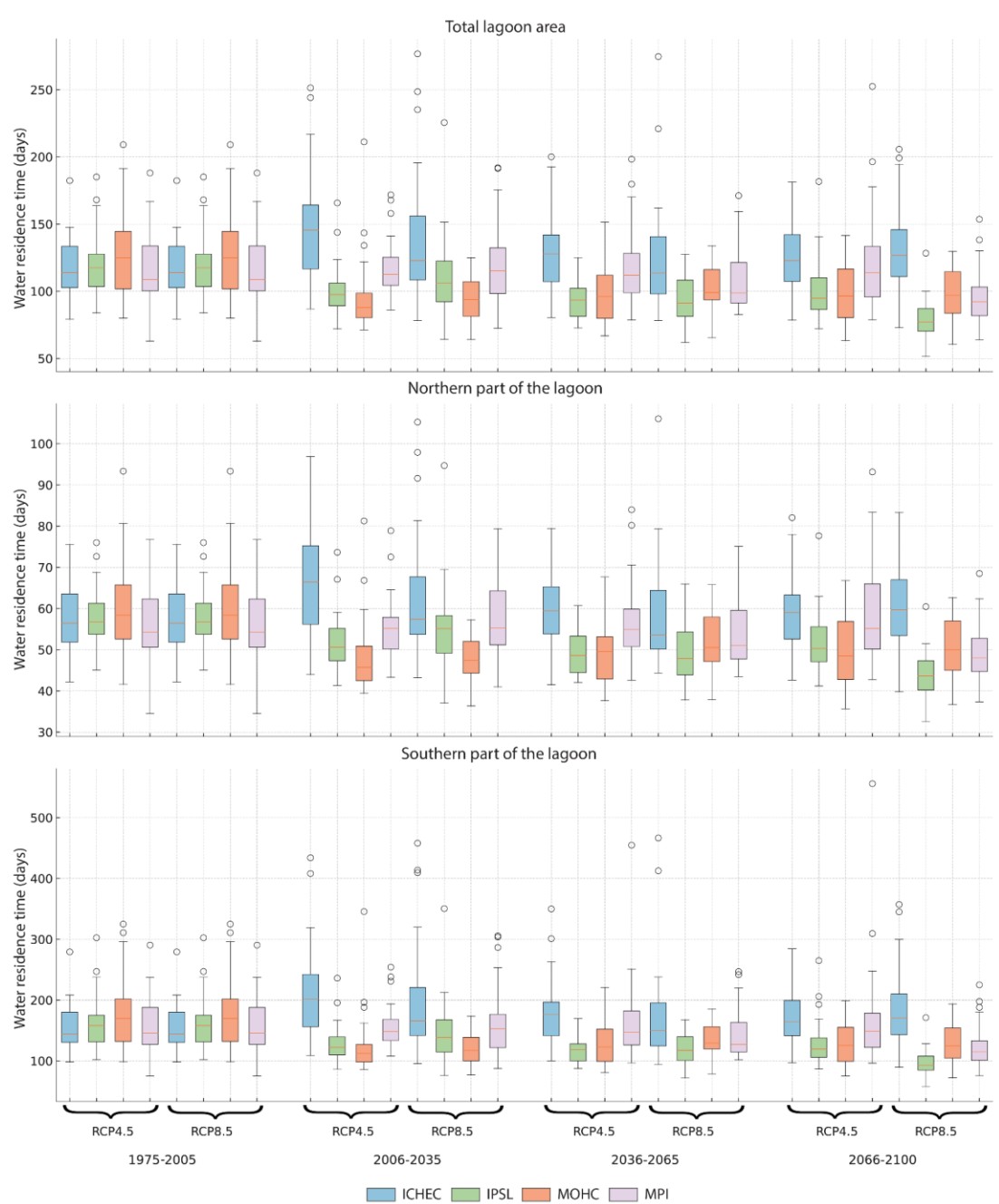

**Figure A2. Average water residence time (in days) in the total lagoon area, northern and southern parts**
**under RCP4.5 (left column) and RCP8.5 (right column) scenarios splitted to 30 years periods.**

**Appendix B: Results of the changepoint analysis**

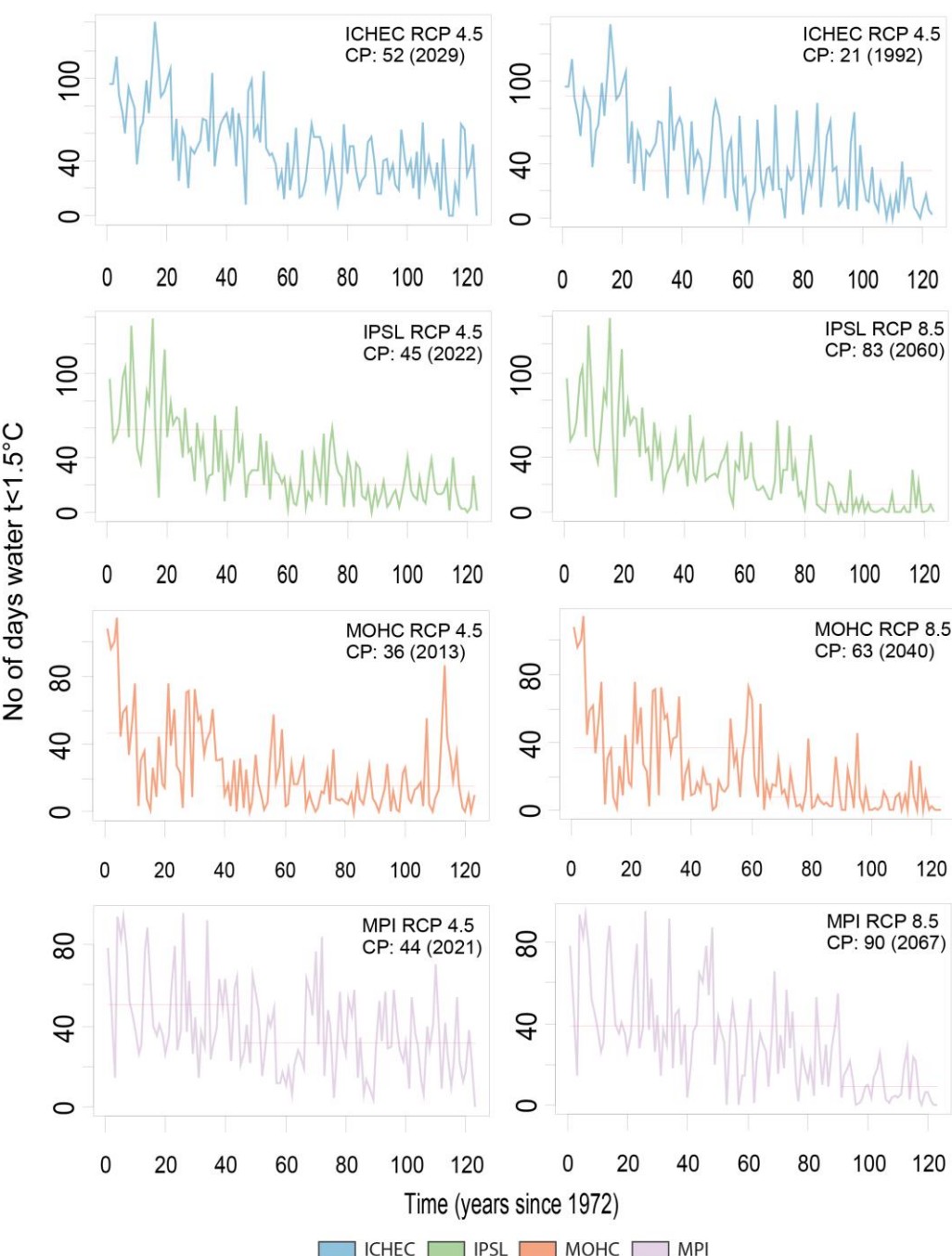


**Figure B1. Single change point (CP) detection in the modeled time series of burbot spawning period t<1.5ºC duration (Vente**
**area). Means (M) and variances (V) of two periods are provided.**

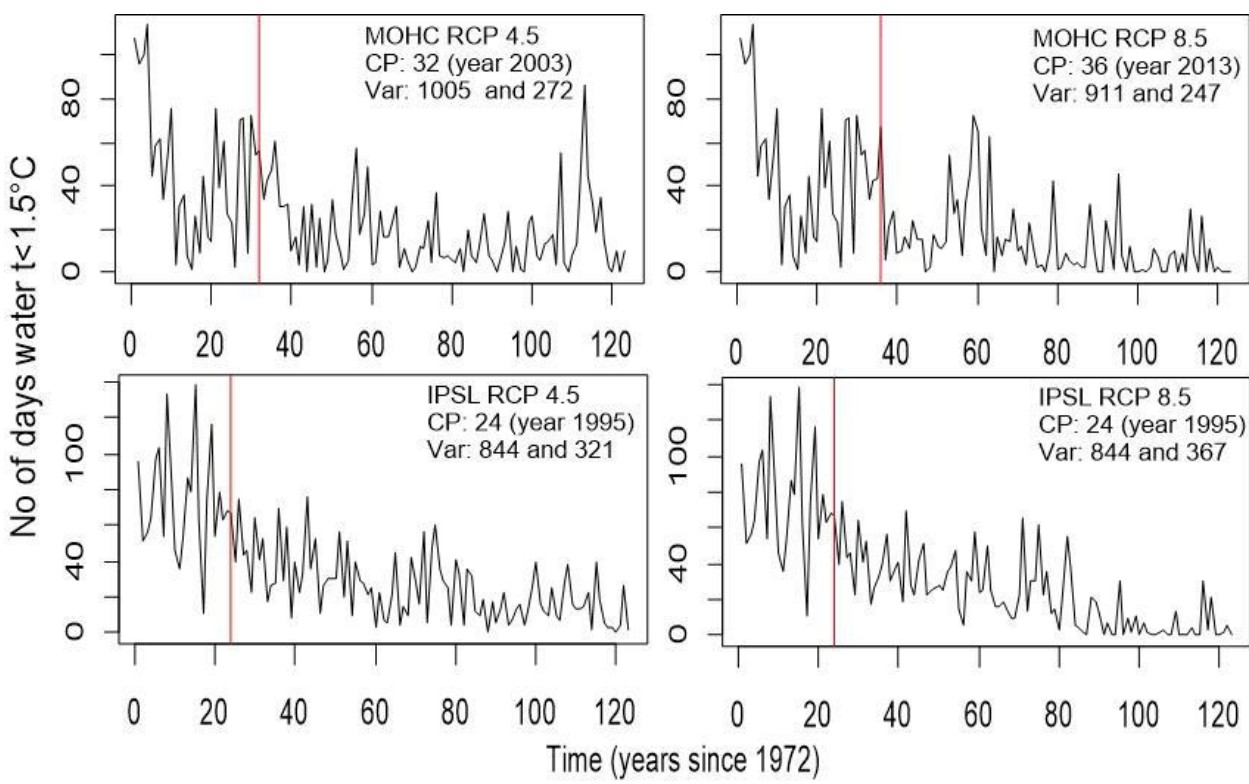


Figure B2. Single change point (CP) of variances detection in the modeled time series of burbot spawning period t<1.5ºC
duration (Vente area). Variances (Var) of two periods are provided.
