# Peer review of "Exploring Variability in Climate Change projections on the"

_EGUsphere, 2024_

## Referee Comment (RC2)

Review of Cerkasova et al., Egusphere

This comprehensive study dealt with future environmental situation of the Curonian Lagoon as impacted by projected climate change affecting itself and its drainage basin by the end of the 21$^{st}$ century. The study is very well written, the methods are mostly appropriate (see one minor point below on flood timing and a point on standard deviations as an uncertainty metrics), the results are of high scientific value (although they partly repeat the findings from the previous study of the authors from 2023). What I particularly like in this study is its multi-dimensional character and looking at environmental situation at a holistic way. Of course, not all possible parameters are considered, but their number is higher than in the majority of comparable studies which I recall.

I do not like the title. It can be drawn from the title that the authors are "modelling uncertainty". I do not think it is the case. They are modelling future environmental conditions and analysing uncertainty. The tile also suggests that the authors are "modelling the impact of uncertainty on a watershed and lagoon". I do not think this wording is correct (is the "impact of uncertainty" really modelled here?). In summary, I suggest to rethink the title.

The authors used a wide range of different analysis and visualisation techniques for different environmental parameters (line plots with moving averages – sometimes with trend lines and sometimes without; box plots – for entire period which does not say anything about direction of change; annual line plots with trend lines; combined annual and moving average line plots; heatmaps; changepoint detection plots). I personally think that the study would benefit from a more consistent method of presenting the results. If there are good reasons why the results for each parameter are presented in a different way, I failed to understand those. On the positive side, it is good that the Figure 10 summarises the results for different parameters in a consistent ways (could the burbot spawning period also be covered here?).

Specific comments:

Line 66 Avoid the term 'forecasting' (here and elsewhere), it is not a synonym of projections

Your narrative in the Introduction would sound more convincing if you started your thread with the specific problems of the Curonian Lagoon ecosystem that require attention, in particular in the context of projected climate change. Then explain that tackling such problems requires integrated modeling frameworks. And only then start with your introduction on climate modeling uncertainty. Otherwise your first mention about „integrated modeling tools" in line 42 seems to come out of the blue.

Line 100 Should be „two" models

Line 113: maybe „main outlets" instead of „two points"?

Line 118: maybe „key physical variables" instead of „all the physics"

Table 1 Consider modifying Table 1 to distinguish between the actual model input data and the reference data that are used for calibration/validation. Besides, weather data seem to be missing.

Line 128: Please mention that you refer to the future conditions here. Also important to mention about the bias correction method.

Line 160: If your spring flood window starts on 1 Feb, then „spring" is maybe not the best term (cold season flood? snowmelt flood?), but this is not so important. More important is that, as I look at Fig. 5, it is clear that in many cases, particularly towards the end of the century, your date of peak flow

occurrence happens on 1 February. I bet that in most of these cases the actual date of peak flow occurs earlier – in January, or maybe even December of the previous year! A more appropriate method for detecting trends in flood timing is by converting dates to angular values, using the circular statistics approaches (see e.g. Bloschl et al. 2017). Both the p values and trend slopes could be affected by this issue (although there is no doubt that regardless of the method, downward trends will prevail).

Line 177: It is not clear to me why for all variables you applied Mann-Kendall trend detection, but for the burbot spawning temperature indicator you dealt with changepoints.

Line 205: It is not clear for which period the data underlying the box plots were aggregated. I can only guess it was done for the entire simulation period 1975-2100. If this is the case, I would recommend to split this long period into one or two 30-year periods. Aren't we mainly interested in climate change signals? You can than still make comparison between climate models, but not so much about their actual values, but projected changes, which in my opinion is more relevant.

Line 237: I think that the sentence "TP loads could eventually fall below the targets" sounds overly optimistic, looking at Fig. 6. All trend slopes are positive for TP loads, it is just that there are some periods for which the loads could fall below the target, but there is not a single case for which the simulated values would be continuously below the target for a longer period.

Fig. 7: Why in this plot you have shown the annual plot lines in addition to the moving averages?

Fig. 10: Shouldn't it be a table? In addition, Theil-Sen slope estimators are given in absolute values, which only allows to compare them between climate models, but not between environmental parameters. There exist simple methods for standardizing Theil-Sen slope (e.g. express it as an average change per decade relative to some "average" value for a given parameter).

Line 305: Section 3.3 I suggest to be more careful with the wording here regarding 'uncertainty'. The authors seem to treat standard deviation calculated from annual values of various environmental parameters as a measure of uncertainty, whereas in fact it just tells us about inter-annual variability. In literature, model spread is a common (although imperfect) metric for quantifying uncertainty. Model spread is quite nicely visible in Figs 3 and 6. For example, in Fig. 6, TN loads under RCP4.5 are characterized with relatively low and almost constant in time model spread. However, under RCP8.5, model spread is growing in time, and by the end of century becomes huge.

Standard deviation is not really a measure of climate model uncertainty – maybe just one of its facets. If standard deviations from two climate models significantly differ, it indirectly indicates that there could be an offset in future projections

One limitation of your analysis of standard deviations is that you include the entire simulation period of 125 years. It would be more meaningful if this long period was divided into shorter periods and comparison was done between them. And again, comparison of the model spread between the periods would be more informative about uncertainty than standard deviation.

Line 384: In your discussion about uncertainties, you should at least mention about one source that was neglected in this study, namely the regional climate models (RCMs). Your results are based on a single RCM, while different RCMs could yield different results, similar to GCMs. Where there any studies for this region which considered ensembles consisting of multiple GCM-RCM model combinations? Was RCM uncertainty component quantified?

Line 466: Shouldn't it be TN here?

---

## Author Response (AR1)

**Author's response**

The manuscript was revised and due to reviewers comments. Detailed answers provided for the reviewers are below and here we present a summary of changes:

- The title was changed avoiding the uncertainty term and rephrased because of the reviewer comments
- Uncertainty, forecast terms were revised and changed to variety, variability and projections.
- Some typing errors were found and corrected.
- An abstract was revised and the sentences about models were clarified.
- Introduction was revised according to reviewer comments, adding the explanation about specific problems of the lagoon ecosystem.
- In the material section we added more details about the modeling system and bias correction methods.
- In section 3.1.2. we expanded the description of spring flood
- Discussion section were revised following the reviewer comments
- All figures were harmonized. The style and colors were changed to be consistent.
    - Fig. 4 were changed from the box plot to line plot. The boxplots were moved to Appendixes
    - In Fig 10 and 11 results about burbot spawning period were added.

**General Comment**:

The authors have resumed their former work to provide further insights in the study of Curonian Lagoon dynamics in a climate scenarios perspective. They have leveraged the modelling setup of the previous study and have stated the aims of their latest study in a clear manner inside a streamlined text. The concept of forcing an hydrodynamic model with accurate information, such as the one coming from an hydrological model, constitutes a better practice with respect to provide climatology derived river water inputs, the latter being deprecated if not, as it might be in some cases, detrimental. Yet some technical aspects of the paper can be ameliorated in two points which most drew my attention:

1. the modelling setup is poorly described in detail. The authors refer to their former paper for the model description, that is not sufficient in that case either in my opinion.

2. the calculation of water fluxes is lacking of some details that deserve to be reported. Please see the comments below for detailed suggestions and offers of reflection.

Authors' response:

Thank you for your thorough review of our manuscript. We have considered your comments and suggestions for improving the technical aspects of our paper. Below is a summary of the revisions we've made based on your specific points:

1. **Modeling Setup Description:** We have expanded the section detailing our modeling setup. We now provide a more detailed description of the hydrodynamic and hydrologic model configuration, including the specific parameters, boundary conditions used in our study (section 2.2.).
2. **Calculation of Water Fluxes:** We have revised the section on the modeling set-up including additional details of water flux calculations. This now covers the methodologies employed and the assumptions made (section 2.2.).

We believe these revisions address your raised concerns and enhance the clarity of our study. Thank you again for your valuable feedback, which has been instrumental in improving the quality of our manuscript.
* * *
lines 38:39 "Apart from the atmospheric models, there is also a variety of ocean models  that have to be considered (Madec et al.,2016, Mellor G. L., 2004, Umgiesser et al. 2004)."

This sentence does not look really coherent (a variety is mentioned but 3 models only are referenced).  The authors should re-formulate the sentence and justify the reference to these 3 specific circulation models.

Authors' response:

Thank you for pointing out the inconsistency in our statement. We agree that the sentence could be more precise. We have revised the sentence to clarify our intent and to justify the specific references:

lines 39-42: Apart from the atmospheric models, there is also a variety of ocean models, for example NEMO (Madec et al., 2016), POM (Mellor, 2004), ROMS (Shchepetkin and  McWilliams, 2005), MITgcm (Marotzke et al., 1999), SHYFEM ( Umgiesser et al. 2004) and others, that have to be considered.

References:
        Madec, G., and NEMO System Team: NEMO ocean engine, Sci. Notes Clim. Model. Cent., 27, ISSN 1288-1619, Institut Pierre-Simon Laplace (IPSL), 2004.
        Marotzke, J., Giering, R., Zhang, K. Q., Stammer, D., Hill, C.,; Lee, T.: Construction of the adjoint MIT ocean general circulation model and application to Atlantic heat transport sensitivity. Journal of Geophysical Research: Oceans, 104, 29529–29547, 1999. https://doi.org/10.1029/1999JC900236
        Mellor, G. L.: Users guide for a three-dimensional primitive equation numerical ocean model, Princeton Univ., Princeton, NJ, 08544–10710, 2004.
        Shchepetkin A.F., McWilliams, J.C.: The regional oceanic modeling system (ROMS): a split-explicit, free-surface,  topography-following-coordinate  oceanic  model.  Ocean  Modelling,9  (4),  347-404,  2005. https://doi.org/10.1016/j.ocemod.2004.08.002.
        Umgiesser, G., Melaku Canu, D., Cucco, A., Solidoro, C.: A finite element model for the Venice Lagoon. Development, up, calibration and https://doi.org/10.1016/j.jmarsys.2004.05.009, 2004.
* * *
lines 72:74

The sentence doesn't really look like well-constructed. I suggest something like:

"The lagoon covers an area of 1584 km2, with its widest section stretching up to 46 km in the southern part. Conversely, in the northernmost part (Klaipėda Strait), it narrows down to approximately 400 m wide."

Authors' response:

Thank you for the remark, we incorporated your suggested sentence reformulation in the updated version of the manuscript (section 2.1 Study area), lines: 77-79
* * *
lines 116:118

The text can be enriched with more details about the shyfem configuration, such as the horizontal resolution, the type of boundary conditions (lateral/surface). Considering the climate context, what kind of interpolation of atmospheric field has been applied to force SHYFEM? What kind of bulk formulation has been applied?

Authors' response:

We added additional sentences to the text, lines 124-129.

Horizontal resolution is variable due to the finite element nature of the grid. However, it varies from 250 m close to the Klaipeda Strait to up to 2.5 km in the central part of the lagoon and up to 10 km in the Baltic Proper. The atmospheric forcing has been interpolated directly from the regular grid of the regional climate model data to the finite element nodes by bi-linear interpolation. Lateral boundary conditions have been taken from Copernicus data and interpolated onto the finite element grid (water levels, T, S). The COARE3.0 module is used for bulk formulation.---
* * *
line 186
Can the authors add more details about the methodology to compute the water fluxes across the section? To this end the authors should address these 2 points:

1. In their previous study (Idzelyte,2023) the authors have split the flow exchange computations in "inflow" and "outflow" in order to assess the variation in percentage in the various scenarios/seasons. Considering that in this study the authors address only fluxes timeseries, does this inflow/outflow distinction applies yet? How is the 10-year moving average computed in this case? When computing outflow in a 10-year window, all the inflow values that fall in the window are set to 0? The author should provide some details about the methodology of computing the time-averaged fluxes.
2. The authors should provide further insights on the computation of water fluxes across the section. In particular they should report their method of assessing the velocity on the straight line represented by the 4 cross sections. For what I can notice from the authors' former publication of 2023, the SHYFEM mesh is not regular in the Curonian lagoon, where the triangles have different size on the coast and in the center. This makes the computation of fluxes in a conservative way quite tricky. The correct way to compute water fluxes on a mesh like SHYFEM's, and considering also the location of SHYFEM's velocities, is along a spline that connects the triangle centers to the mid-edge points. Computing fluxes accurately across straight segments like the 4 proposed by the authors is possible but upon the application of a conservative method of interpolation on the velocity field. Have the authors considered this issue and its possible effect on the uncertainty assessment?

Authors' response:

1. Yes, the distinction between inflow and outflow categories still applies in this analysis. The 10-year moving average is computed by first calculating the yearly average flux for both inflow and outflow categories separately. These yearly averages are then used to compute the moving average over a 10-year window. For each 10-year window, we calculate the average inflow and outflow flux by considering all yearly-averaged values within that period. The inflow values are not set to zero when computing the outflow, and vice versa. Instead, both inflow and outflow are continuously accounted for over the entire period to ensure accuracy in reflecting the overall water flux dynamics. We have included a detailed explanation of this methodology in the revised manuscript to clarify the process for computing the time-averaged fluxes. The changes (as written below) can be found in the updated manuscript section "2.4.1 Investigation of hydrological and hydrodynamic model results":

*"In this analysis, we maintained the inflow and outflow categories as in our previous study (Idzelytė et al., 2023). We analyzed the data by computing the 10-year moving average using yearly*

*average fluxes, this way ensuring an accurate representation of water flux dynamics throughout the study period."*

Reference:
    Idzelytė, R., Čerkasova, N., Mėžinė, J., Dabulevičienė, T., Razinkovas-Baziukas, A., Ertürk, A., and Umgiesser, G.: Coupled hydrological and hydrodynamic modeling application for climate change impact assessment in the Nemunas river watershed–Curonian Lagoon–southeastern Baltic Sea continuum, Ocean Sci., 19(4), 1047–1066, https://doi.org/10.5194/os-19-1047-2023, 2023.

2.      To compute the water fluxes across the sides of the elements, first the conservation of mass in the finite volume around a node that is guaranteed by the continuity equation is used. The fluxes over the lines delimiting the finite volume element per element (a split line that connects the triangle centers to the mid-edge points) are made divergence free by subtracting the storage of water inside the node. With these finite volume fluxes the fluxes over the element sides can be computed. In case of the presence of a material boundary the matrix that connects the finite volume fluxes to the fluxes over the element sides is regular and can be solved directly. However, in case of no material boundary the matrix is singular. In this case one of the flux conservation equations is dropped (it is redundant) and is substituted by a condition of non-rotational flow around the node. This procedure ensures a complete mass conservation over the lines defined on the element sides and is correct up to machine precision.

 To the manuscript we will add: "To compute the water fluxes across the sides of the elements, first the conservation of mass in the finite volume around a node that is guaranteed by the continuity equation is used. The fluxes over the lines delimiting the finite volume element per element are made divergence free by subtracting the storage of water inside the node. With these finite volume fluxes the fluxes over the element sides are computed."
* * *
Caption of Fig.3 "Note the adjusted y-axis ranges"

Authors' response:

Thank you for bringing up this oversight. We have made the necessary corrections in this caption.
* * *
Section 3.1.6

As I understand the authors have used an ice model to force the simulations but I cannot find any reference (I suppose is Tedesco et al. 2009, please add it) nor how it's been nested in the modeling framework. It is not clear whether ESIM2 has forced SHYFEM or it has been used as standalone. I think that the modelling framework description paragraph should be more exhaustive.

Authors' response:

Yes, the ice thickness data were derived using the ESIM2 model as presented by Tedesco et al., 2009. The ESIM2 model was operated independently as a standalone model. Its output time series were subsequently integrated as surface boundary input data for the hydrodynamic component of our modeling system. We have now included a detailed explanation of this process in the revised manuscript to clarify the modeling framework. You can find the following updated text in section "2.3 Data":

*"The ice thickness data utilized in our study were derived using the ESIM2 model (Tedesco et al., 2009, Idzelytė and Umgiesser, 2021). This model was run independently as a standalone system, and the resulting output time series were integrated into our hydrodynamic modeling framework*

*as surface boundary input data. This approach allowed us to accurately incorporate ice thickness dynamics into our simulations, enhancing the overall reliability of our model during the ice season."*

References:
Tedesco, L., Vichi, M., Haapala, J., and Stipa, T.: An enhanced sea-ice thermodynamic model applied to the Baltic Sea, Boreal Environ. Res., 14, 68–80, 2009.
Idzelytė, R. and Umgiesser, G.: Application of an ice thermodynamic model to a shallow freshwater lagoon, Boreal Environ. Res., 26, 61–77, 2021.

Technical corrections:

In addition, we correct some small typing errors found after additional proofreading. And we revised our manuscript a lot, based on the other reviewer comments.

**Reviewer 2**

Review of Cerkasova et al., Egusphere
This comprehensive study dealt with future environmental situation of the Curonian Lagoon as impacted by projected climate change affecting itself and its drainage basin by the end of the 21st century. The study is very well written, the methods are mostly appropriate (see one minor point below on flood timing and a point on standard deviations as an uncertainty metrics), the results are of high scientific value (although they partly repeat the findings from the previous study of the authors from 2023). What I particularly like in this study is its multi-dimensional character and looking at environmental situation at a holistic way. Of course, not all possible parameters are considered, but their number is higher than in the majority of comparable studies which I recall.

First, we thank the reviewer for the valuable comments and remarks. We revised our manuscript and answered the reviewers questions.

I do not like the title. It can be drawn from the title that the authors are "modelling uncertainty". I do not think it is the case. They are modelling future environmental conditions and analysing uncertainty. The tile also suggests that the authors are "modelling the impact of uncertainty on a watershed and lagoon". I do not think this wording is correct (is the "impact of uncertainty" really modelled here?). In summary, I suggest to rethink the title.

Authors' response:

Noting the reviewers comments, we agree to change the title and reformulate it as:
Exploring Variability in Climate Change projections on the Nemunas River and Curonian Lagoon: coupled SWAT and SHYFEM modeling approach

The authors used a wide range of different analysis and visualisation techniques for different environmental parameters (line plots with moving averages – sometimes with trend lines and sometimes without; box plots – for entire period which does not say anything about direction of change; annual line plots with trend lines; combined annual and moving average line plots; heatmaps; changepoint detection plots). I personally think that the study would benefit from a more consistent method of presenting the results. If there are good reasons why the results for each parameter are presented in a different way, I failed to understand those. On the positive

side, it is good that the Figure 10 summarises the results for different parameters in a consistent ways (could the burbot spawning period also be covered here?).

Authors' response:
When performing the analysis, we were targeting stakeholders and the information which could be used later for the ecological evaluation of the system.
Since the graphs are hardly readable due to large dataset count, we added the trend lines where the dataset was less condense and the trend was readable in the graph. The plots with 10-year moving average were chosen for the parameters with high variability between the years and were challenging to read otherwise.
We agree that the data in a box-plot does not show the direction of change, however the direction of change is summarized in Figure 10. The numbers of the averaged water residence time are important for the ecological evaluation and we hope for higher citation of these data. However, we value the reviewer's comments and due to his concerns we have changed the boxplot picture Fig. 4 to a line plot to be more consistent. Furthermore, we moved the boxplots to the Annexes to keep this information in a manuscript.
The figure types were chosen to best represent the variation in the projections for specific outputs. We harmonized the data representation by using the same color palette so that the reader would see consistency in between the figures. We changed Fig.7 and 8, to have the same colors for scenarios as in previous pictures.
According to burbot analysis we will add requested data to Fig. 10.
Although this might seem like a diverse visualization technique, we'd prefer to keep the current and updated figure types.

Specific comments:
Line 66 Avoid the term 'forecasting' (here and elsewhere), it is not a synonym of projections

Authors' response:

Thank you for the remark, we have revised the manuscript accordingly.

Your narrative in the Introduction would sound more convincing if you started your thread with the specific problems of the Curonian Lagoon ecosystem that require attention, in particular in the context of projected climate change. Then explain that tackling such problems requires integrated modeling frameworks. And only then start with your introduction on climate modeling uncertainty. Otherwise your first mention about „integrated modeling tools" in line 42 seems to come out of the blue.

Authors' response:

We revised the introduction based on your comments. We believe that the introduction is more convincing and clear now.

Line 100 Should be „two" models
Line 113: maybe „main outlets" instead of „two points"?
Line 118: maybe „key physical variables" instead of „all the physics"

Authors' response:

Thank you for your detailed remarks. We made the mentioned revisions in the updated version of the manuscript.

Table 1 Consider modifying Table 1 to distinguish between the actual model input data and the reference data that are used for calibration/validation. Besides, weather data seem to be missing.

Authors' response:

Missing information is added to Table 1.

Line 128: Please mention that you refer to the future conditions here. Also important to mention about the bias correction method.

Authors' response:

We added more details about the bias correction and referred to future conditions in Section 2.3.

Line 160: If your spring flood window starts on 1 Feb, then „spring" is maybe not the best term (cold season flood? snowmelt flood?), but this is not so important. More important is that, as I look at Fig. 5, it is clear that in many cases, particularly towards the end of the century, your date of peak flow occurrence happens on 1 February. I bet that in most of these cases the actual date of peak flow occurs earlier – in January, or maybe even December of the previous year! A more appropriate method for detecting trends in flood timing is by converting dates to angular values, using the circular statistics approaches (see e.g. Bloschl et al. 2017). Both the p values and trend slopes could be affected by this issue (although there is no doubt that regardless of the method, downward trends will prevail).

Authors' response:
Regarding the "spring flood" - a good observation, however this term is used by the local stakeholders. As we target them, we will keep the term, and add a clarification to the reader, why this particular period was chosen and why it is called "spring flood" (first lines in paragraph 3.1.2.).
"The high discharge of the Nemunas River and subsequent flooding of the delta region is a nearly annual event which occurs in late winter - spring season, and is referred to as "spring flood" in Lithuania. We use the same term in this study and consider the historic period of high river flows to be from 1st of February to 30th of April."

Regarding the flood timing - it is also a very good observation and suggestion! We will definitely include this in our future studies. For this paper, as we target a specific situation, we will keep the analysis and the results. The paper text is now amended to reflect this drawback and potential future work (last sentences in section 4.1).
"Moreover, the projections show that the timing of spring high flows are moved to the boundary of the analyzed period (February 1st), which indicates that the peak flow rate might occur even earlier. Although not analyzed in this paper, a follow-up study will explore these projections using more appropriate methods for detecting trends in flood timing, i.e. using the circular statistics approaches (Bloschl et al., 2017)."

Line 177: It is not clear to me why for all variables you applied Mann-Kendall trend detection, but for the burbot spawning temperature indicator you dealt with changepoints.

Authors' response:

The trend of annual temperatures shows a clear tendency for the increase, i.e. the gradual long-term change from past conditions. However, the modeling data also show some abrupt changes, especially at the beginning of the modeled time series in 90's. For the conservation purposes it is more important to detect those abrupt changes in the system and relate them to stock data, and fisheries practices. Identifying the further change points when the spawning season becomes

critically short, it is possible to inform fishery managers about necessary measures for species protection. To sum up, changepoint detection is a more beneficial statistical indicator for the possible stakeholders (fishermen and relevant policy authorities). Nevertheless, to make the findings systematic, we add the trend analysis results to Fig. 10.

Line 205: It is not clear for which period the data underlying the box plots were aggregated. I can only guess it was done for the entire simulation period 1975-2100. If this is the case, I would recommend to split this long period into one or two 30-year periods. Aren't we mainly interested in climate change signals? You can than still make comparison between climate models, but not so much about their actual values, but projected changes, which in my opinion is more relevant.

Authors' response:

We already made the changes based on the previous comment. We changed the box plots to line plots, and moved the box plot to Appendix A. However, we agree with the reviewer that the figure splitted in a 30 year period could complement the analysis, therefore we added this picture to Appendix A as well.

[Figure]

Fig. 4. Will be plotted as lines and the box plots moved to Appendixes

Line 237: I think that the sentence "TP loads could eventually fall below the targets" sounds overly optimistic, looking at Fig. 6. All trend slopes are positive for TP loads, it is just that there are some periods for which the loads could fall below the target, but there is not a single case for which the simulated values would be continuously below the target for a longer period.

Authors' response:
That is an astute observation. We agree and rephrase the corresponding text to reflect a more in-depth analysis. Now the statement reads: "TP loads can fall below the maximum target during several brief periods, but the timing of this will depend on the actual climate scenario that unfolds."

Fig. 7: Why in this plot you have shown the annual plot lines in addition to the moving averages?

Authors' response:

This was done to display the underlying variability. While the moving averages help to smooth out short-term fluctuations and highlight longer-term trends, the annual plot lines provide a clear view of the year-to-year variability in the data. This combination allows for a more comprehensive understanding of both the general trend and the fluctuations that occur on an annual basis, offering a fuller picture of the data's behavior over time.

Fig. 10: Shouldn't it be a table? In addition, Theil-Sen slope estimators are given in absolute values, which only allows to compare them between climate models, but not between environmental parameters. There exist simple methods for standardizing Theil-Sen slope (e.g. express it as an average change per decade relative to some "average" value for a given parameter).

Authors' response:

Since the journal does not allow the colored cell tables, we call it a figure. The table with numbers only does not show a full window off the analysis, as a result, we would like to leave it as it is. The manuscript covers many parameters between the models and climate projections, therefore additional analysis (important and interesting it may be) is out of the scope of this paper. In addition, we have ideas for the future publication, where other methods will be used to evaluate results from climate simulations, including above mentioned (i.e. flood timing, splitting of the projection period, etc.).

Line 305: Section 3.3 I suggest to be more careful with the wording here regarding 'uncertainty'. The authors seem to treat standard deviation calculated from annual values of various environmental parameters as a measure of uncertainty, whereas in fact it just tells us about inter-annual variability. In literature, model spread is a common (although imperfect) metric for quantifying uncertainty. Model spread is quite nicely visible in Figs 3 and 6. For example, in Fig. 6, TN loads under RCP4.5 are characterized with relatively low and almost constant in time model spread. However, under RCP8.5, model spread is growing in time, and by the end of century becomes huge.
Standard deviation is not really a measure of climate model uncertainty – maybe just one of its facets. If standard deviations from two climate models significantly differ, it indirectly indicates that there could be an offset in future projections
One limitation of your analysis of standard deviations is that you include the entire simulation period of 125 years. It would be more meaningful if this long period was divided into shorter periods and comparison was done between them. And again, comparison of the model spread between the periods would be more informative about uncertainty than standard deviation.

Authors' response:

It is true that we did not follow Moss and Schneider, 2000; Manning et al., 2004; Mastrandrea et al., 2010 for uncertainty evaluation. To make the article more clear, we rename the section to "Variability in the projections". We reformulated this Section 3.3. and the rest of the manuscript and changed the uncertainty terms to variations or variability, where applicable.

In section 3.3 we discuss the standard deviation and variation of different models and we agree that these statistics only partly explain the uncertainty. The manuscript was prepared from the project's final results, which were carried out according to the project proposal and targeted at specific stakeholders, as mentioned before. Nevertheless, we agree that for a scientific paper it may be a bit confusing. We hope that this revised version of the text will clear this confusion.

Line 384: In your discussion about uncertainties, you should at least mention about one source that was neglected in this study, namely the regional climate models (RCMs). Your results are based on a single RCM, while different RCMs could yield different results, similar to GCMs. Where there any studies for this region which considered ensembles consisting of multiple GCM-RCM model combinations? Was RCM uncertainty component quantified?

Authors' response:

For consistency, we used the same forcing source for both models (SHYFEM and SWAT), and only the RCA4 met all the data set needs for both model setups. No other RCM provided such overlap of forsings and/or area. It is common that the Baltic Sea studies are carried out based on this model only (i.e. Huttunen et al., 2021; Rusu, 2020;Soomere, 2022; Bonaduce et al., 2019). There are studies, where different RCM were used for SWAT model (i.e. Plunge et al. 2022 and

2023), these studies are cited and are in line with our study results. This is a valuable comment, and we add a remark in the text addressing this concern.

References:

Blöschl G. et al. Changing climate shifts timing of European floods. Science 357, 588-590. DOI:10.1126/science.aan2506. 2017.

Bonaduce, A., Staneva, J., Behrens, A., Bidlot, J.-R., and Wilcke, R. A. I.: Wave climate change in the North Sea and Baltic Sea, J. Mar. Sci. Eng., 7, 166, https://doi.org/10.3390/jmse7060166, 2019.

Huttunen, I., Hyytiäinen, K., Huttunen, M., Sihvonen, M., Veijalainen, N., Korppoo, M., and Heiskanen, A.-S.: Agricultural nutrient loading under alternative climate, societal and manure recycling scenarios, Sci. Total Environ., 783, 146871, https://doi.org/10.1016/j.scitotenv.2021.146871, 2021.

Plunge S, Gudas M, Povilaitis A, Piniewski M. Evaluation of the costs of agricultural diffuse water pollution abatement in the context of Lithuania's water protection goals and climate change. Environ Manage. 2023 Apr;71(4):755-772. doi: 10.1007/s00267-022-01745-1. Epub 2022 Nov 11. PMID: 36369297; PMCID: PMC10017570.

Plunge S, Gudas M, Povilaitis A. Effectiveness of best management practices for non-point source agricultural water pollution control with changing climate – Lithuania's case. Agr Water Manag. 2022;267:107635. doi: 10.1016/j.agwat.2022.107635.

Rusu, E.: An evaluation of the wind energy dynamics in the Baltic Sea, past and future projections, Renew. Energy, 160, 350–362, https://doi.org/10.1016/j.renene.2020.06.152, 2020.

Soomere, T.: Numerical simulations of wave climate in the Baltic Sea: a review, Oceanologia, 65, 117–140, https://doi.org/10.1016/j.oceano.2022.01.004, 2023.

Line 466: Shouldn't it be TN here?

Authors' response:

Yes, thank you. It is corrected. Now 544: "However, a severe discrepancy from the targeted loads of TP is projected forecasted by the middle of the century by all models and especially by MOHC"

In addition, we correct some small typing errors found after additional proofreading.